# Online Contract Design With Unknown Technology

**Matteo Bollini** [1]   **Matteo Castiglioni** [1]   **Alberto Marchesi** [1]

## Abstract

*Hidden-action principal-agent problems* model scenarios in which a principal induces an agent to take a costly and *unobservable* action through the provision of outcome-dependent payments. These problems find application in a variety of real-world settings, such as crowdsourcing, online labor platforms, and machine learning task delegation. Recently, much of the literature has focused on how to handle the principal's *uncertainty* about the agent and the surrounding environment, which is often the main challenge in practice. One prominent approach is to adopt an *online learning* framework, where the principal repeatedly interacts with the agent to learn optimal payments from experience. However, existing learning algorithms, while achieving regret that scales sublinearly in the number of interaction rounds $T$, typically suffer from an exponential dependence on the size of the problem instance. In this paper, we show that this problematic exponential growth can be avoided by assuming that the principal has knowledge of a set of possible actions of the agent, while remaining unaware of which actions are actually available—an assumption that is reasonable in many real-world settings.

## 1. Introduction

*Hidden-action principal-agent problems* (Grossman & Hart, 1983) model the interaction between a principal and an agent, where the latter takes an action that induces externalities on the former. Specifically, the agent incurs a cost $c_a$ for taking action $a$, which stochastically determines an outcome $\omega$ associated with a principal's reward $r_\omega$, according to an action-specific probability distribution $F_a$. The central challenge of these problems is that the action chosen by the agent is *unobservable* to the principal. Therefore, the only

[1]Dipartimento di Elettronica, Informazione e Bioingegneria, Politecnico di Milano, Milan, Italy. Correspondence to: Matteo Bollini <matteo.bollini@polimi.it>.

*Proceedings of the 43rd International Conference on Machine Learning*, Seoul, South Korea. PMLR 306, 2026. Copyright 2026 by the author(s).

way in which the principal can induce the agent to take a favorable action is through an outcome-dependent payment scheme $p$, called a *contract*, which specifies a payment $p_\omega$ from the principal to the agent for each possible outcome $\omega$. Under a contract $p$, the agent selects a *best response* $a(p)$, namely an action $a$ that maximizes the agent's expected utility $\mathbb{E}_{\omega \sim F_a}[p_\omega - c_a]$. The principal's goal is to *commit to an optimal contract*, which is a contract $p$ that maximizes the principal's expected utility $\mathbb{E}_{\omega \sim F_{a(p)}}[r_\omega - p_\omega]$.

Nowadays, hidden-action principal-agent problems find application in a variety of real-world settings, such as crowdsourcing (Ho et al., 2014), online labor platforms (Kaynar & Siddiq, 2023), blockchain (Cong & He, 2019), delegation of machine learning tasks (Saig et al., 2023), and pay-for-performance healthcare (Bastani et al., 2016; 2019). Moreover, algorithmic contract design is playing an increasingly important role in today's world, which relies more and more on AI agents to perform complex tasks (see, *e.g.*, (Hadfield-Menell & Hadfield, 2019; Saig et al., 2024)).

Recently, considerable efforts have been devoted to studying the computational aspects of hidden-action principal-agent problems (see, *e.g.*, (Dütting et al., 2024)). Classic approaches assume that the principal *knows everything* about the environment (see, *e.g.*, (Laffont & Martimort, 2002)). However, this assumption considerably limits practical applicability, as in real-world problems typically little is known about the surrounding environment. Thus, the focus has recently shifted toward settings in which some parameters of the environment are *unknown*. Some works (see, *e.g.*, (Guruganesh et al., 2021; Alon et al., 2021; Castiglioni et al., 2022a)) studied *Bayesian* settings in which the principal is uncertain about the agent's actions, whose costs and outcome distributions are assumed to be determined by the agent's private type drawn from a *known* probability distribution. Other works (see, *e.g.*, (Carroll, 2015; Walton & Carroll, 2022; Bernasconi et al., 2024)) took a *worst-case* approach to uncertainty, without making any assumptions on the distribution of the agent's types. Finally, some works (see, *e.g.*, (Zhu et al., 2023; Han et al., 2024; Bacchiocchi et al., 2024)) modeled uncertainty by casting principal-agent problems into *online learning* frameworks.

This paper follows the online learning approach. We study hidden-action principal–agent problems in which a principal

repeatedly interacts with an agent over $T$ rounds. At each round $t$, the principal commits to a contract $p^t$, and the agent chooses a best response $a^t$. The principal then observes the realized outcome $\omega^t \sim F_{a^t}$ and collects a utility of $r_{\omega^t} - p_{\omega^t}$. As is customary in online learning, the goal of the principal (the learner) is to minimize the *regret*, defined as the difference between the utility obtained by always committing to an optimal contract and the utility actually achieved over the $T$ rounds. Zhu et al. (2023) initiated the study of these online learning problems by providing a tight regret bound with dependence on $T$ of order $T^{1-\Theta(1/d)}$, where $d$ is the number of outcomes. Subsequently, Bacchiocchi et al. (2024) attempted to circumvent the problematic exponential dependence of the regret on $d$. Specifically, they achieve a regret of the order of $d^n T^{4/5}$, where $n$ is the number of agent's actions. While this scales much better in $T$, the regret still suffers from an exponential dependence on $n$.

In this paper, we investigate the following question:

*"Is it possible to design online learning algorithms that achieve regret growing sublinearly in $T$ without exhibiting an exponential dependence on the instance size?"*

### 1.1. Original Contributions

We answer the question above affirmatively. Specifically, we show that by introducing a mild assumption on the principal's knowledge of the agent, it is possible to design an online learning algorithm that achieves regret growing as $T^{4/5}$, while simultaneously scaling polynomially in the parameters characterizing the size of the problem instance.

We consider the case in which the principal does *not* know the agent's actual *technology*—the set of actions truly available to the agent—but only knows a superset $A$ of it. This assumption is reasonable in many real-world settings, where the principal typically has a general understanding of all potential choices an agent might have, but does not know which ones are actually feasible. For instance, when a client delegates a machine learning task to a provider, the client may know all the model/training choices potentially available to the provider, but *not* those that are actually feasible for them due to private constraints.

A naïve approach to tackle the problem would be to instantiate a classic online learning algorithm (such as, *e.g.*, UCB1 (Auer et al., 2002)) on an equivalent multi-armed bandit instance, with one arm for each possible subset of $A$. While this would yield regret growing sublinearly in $T$, it incurs an exponential dependence on the number of possible actions. Moreover, it suffers from an exponential per-round running time, due to the exponential number of arms that must be considered. We take a completely different approach, which aims to directly learn the agent's actual technology $A^\star \subseteq A$. The main challenge is that, with only the realized outcome

$\omega^t$ as feedback, the set $A^\star$ is unlearnable in a reasonable number of rounds: there may exist two actions whose outcome distributions are extremely similar and thus difficult to distinguish. We circumvent this issue by working with *meta-actions*, which are subsets of actions $M \subseteq A$ that have similar distributions. Our approach can be conceptually split into two phases. In the *first* phase, the goal is to identify which meta-actions are actually existing, namely those $M$ such that $M \cap A^\star \neq \varnothing$, and, for each of them, to characterize the region of the contract space in which its actions are best responses for the agent. Since these regions are *not* polytopes, we learn suitable polytopal approximations of them. Then, in the *second* phase, we commit to an approximately optimal contract for the remaining rounds.

### 1.2. Related Works

Online learning in hidden-action principal-agent problems has received increasing attention in recent years (Ho et al., 2014; Cohen et al., 2022; Zhu et al., 2023; Dütting et al., 2023; Bacchiocchi et al., 2024; Chen et al., 2024; Han et al., 2024; Bacchiocchi et al., 2025a;b). In particular, the online learning framework studied in this paper was first introduced by Zhu et al. (2023), who provided an algorithm with regret $\widetilde{\mathcal{O}}(\sqrt{d} \cdot T^{1-1/(2d+1)})$, where $d$ is the number of outcomes. Such an exponential dependence on $d$ is unavoidable in general. Bacchiocchi et al. (2024) later showed that this barrier can be circumvented when the number of the agent's actions $n$ is small. Specifically, they provide an algorithm with regret of order $d^n T^{4/5}$. Our algorithm avoids any exponential dependence on the instance size, while retaining the $T^{4/5}$ dependence on $T$ of (Bacchiocchi et al., 2024).

Some works (Zhu et al., 2023; Bacchiocchi et al., 2025a) also consider a simpler version of the online learning framework studied in this paper, in which the principal is restricted to committing only to *linear* contracts, namely payment schemes parameterized by a single scalar specifying the fraction of the principal's reward given to the agent. In such a setting, Zhu et al. (2023) show that the regret grows as $\Theta(T^{2/3})$, while Bacchiocchi et al. (2025a) show that a regret of $\widetilde{\mathcal{O}}(\sqrt{nT})$ can be attained when the number of agent's actions $n$ is small relative to $T$, namely when $n \leq T^{1/3}$.

Finally, some works adopt different approaches to handling the principal's uncertainty, such as those studying *Bayesian* settings with distributional assumptions over the agent's type (Guruganesh et al., 2021; 2023; Alon et al., 2021; 2023; Castiglioni et al., 2022a;b; 2023), as well as those taking a *worst-case* approach against all possible agent types (Carroll, 2015; 2019; Walton & Carroll, 2022; Bernasconi et al., 2024; Yu & Kong, 2020; Dütting et al., 2019; Bacchiocchi et al., 2025b). In particular, Carroll (2015) assumes to know a subset of the agent's technology, and proves that linear contracts are max-min optimal in that case.

## 2. Preliminaries

### 2.1. Hidden-Action Principal-Agent Problems

A hidden-action principal-agent problem is characterized by a finite set of $d$ outcomes $\Omega := \{\omega_1, \ldots, \omega_d\}$, a vector $r \in [0,1]^d$ defining a principal's reward $r_\omega \in [0,1]$ for each outcome $\omega \in \Omega$, and a finite set of agent's actions $A^\star$. An action $a \in A^\star$ has a cost $c_a \in [0,1]$ for the agent and results in a (random) outcome $\omega \sim F_a$, where $F_a \in \Delta(\Omega)$ is a probability distribution over outcomes that characterizes action $a$.[1] We denote with $F_{a,\omega} \in [0,1]$ the probability of outcome $\omega \in \Omega$ when the agent chooses action $a \in A^\star$.

The principal's objective is to steer the agent's decision toward a favorable action, without directly observing the action actually selected by the agent. To this end, the principal commits to a *contract*, defined as an outcome-based payment scheme. Formally, a contract is a vector $p \in \mathbb{R}_{\geq 0}^d$ defining a payment $p_\omega \geq 0$ from the principal to the agent for each outcome $\omega \in \Omega$.[2] Under a contract $p \in \mathbb{R}_{\geq 0}^d$, the agent selects an action that is (i) *incentive compatible* (IC), *i.e.*, it maximizes the agent's expected utility, and (ii) *individually rational* (IR), *i.e.*, it yields non-negative expected utility. For ease of presentation, we assume w.l.o.g. that there always exists an action $a \in A^\star$ with cost $c_a = 0$. This allows us to focus solely on incentive compatibility, since any IC action is also IR. Moreover, as is standard in the literature (see, *e.g.*, (Zhu et al., 2023; Bacchiocchi et al., 2024)), we assume bounded payments, *i.e.*, $p \in \mathcal{P} := [0,1]^d$.[3]

We denote by $u^P(p,a)$ (respectively, $u^A(p,a)$) the expected utility of the principal (respectively, the agent) when the principal commits to a contract $p \in \mathbb{R}_{\geq 0}^d$ and the agent chooses an action $a \in A^\star$. Formally, it holds:

$$u^P(p,a) := \langle r - p, F_a \rangle, \quad u^A(p,a) := \langle p, F_a \rangle - c_a.$$

We denote by $a(p)$ the IC action selected by the agent, *i.e.*, $a(p) \in \arg\max_{a \in A^\star} u^A(p,a)$, where we assume that ties are broken in favor of the principal. Thus, the principal's expected utility under contract $p$ is $u^P(p) := u^P(p, a(p))$. The action $a(p)$ is sometimes also referred to as the agent's *best response* under contract $p$.

A principal's optimal contract is $p^\star \in \arg\max_{p \in \mathcal{P}} u^P(p)$. We denote by $a^\star := a(p^\star)$ the best response selected by the agent under such an optimal contract, and by $\mathrm{OPT} := u^P(p^\star)$ the resulting principal's expected utility.

---

[1]We denote with $\Delta(X)$ the set of all the probability distributions over a finite set $X$, *i.e.*, the $(|X|+1)$-dimensional simplex.

[2]The assumption that $p_\omega \geq 0$ (*i.e.*, payments can only be from the principal to the agent) is common in the literature, where it is known as *limited liability* (Carroll, 2015).

[3]The assumption of bounded payments is unavoidable, as otherwise an optimal contract would be unlearnable (Bacchiocchi et al., 2024). We assume payments to lie in $[0,1]$ for ease of presentation, but all our results generalize to the case $[0,B]$ for any $B > 1$.

### 2.2. Learning With Unknown Agent's Technology

We consider an online learning problem in which a principal repeatedly interacts with an agent over $T \in \mathbb{N}$ rounds. At each round $t \in [T]$, the principal commits to a contract $p^t \in \mathcal{P}$. The agent then observes $p^t$ and selects an IC action $a^t := a(p^t)$, which is hidden from the principal. The principal only observes the realized outcome $\omega^t \sim F_{a^t}$ and collects a utility value equal to $u^t := r_{\omega^t} - p_{\omega^t}^t$.

We assume that the principal does *not* know the set of the agent's *existing* actions $A^\star$, which is referred to as the agent's *technology* (Carroll, 2015). Instead, they know a superset of the agent's technology $A \supseteq A^\star$, with $K := |A|$, consisting of all possible agent *candidate actions*, together with their corresponding costs $c_a \in [0,1]$ and distributions $F_a \in \Delta(\Omega)$, for each $a \in A$. Furthermore, we assume that the principal knows the reward vector $r \in [0,1]^d$. For the ease of exposition, we assume w.l.o.g. that every action $a \in A^\star$ is IC under some contract $p \in \mathcal{P}$.

Our goal is to design learning algorithms that commit to a sequence of contracts $p^t \in \mathcal{P}$, for $t \in [T]$, so as to minimize the *(expected) regret* $R_T$ with respect to an optimal contract $p^\star$ over $T$ rounds. Formally, the regret is defined as:

$$R_T := T \cdot \mathrm{OPT} - \mathbb{E}\left[\sum_{t \in [T]} u^t\right],$$

where the expectation is taken w.r.t. the stochasticity of outcomes and the randomization of the algorithm. Ideally, we would like learning algorithms such that: (i) their regret $R_T$ grows polynomially in the size of the problem instance, *i.e.*, in $d$ and $K$, and sublinearly in the number of rounds $T$, and (ii) their per-round running time is polynomial in $t$, $d$, and $K$. When an algorithm satisfies (i), we call it *no-regret*, whereas when it satisfies (ii), we say that it is *efficient*.

### 2.3. Best-Response Sets

We conclude the section by introducing some additional concepts needed in the remaining of the paper.

We denote by $P(a) \subseteq \mathcal{P}$ the *best-response region* of an existing action $a \in A^\star$, which is the set of all the contracts $p \in \mathcal{P}$ under which action $a$ is IC. This region is a polytope defined as the intersection of $[0,1]^d$ with $K-1$ half spaces $\mathcal{H}_{a,a'} \subseteq \mathbb{R}_{\geq 0}^d$, one for each $a \neq a' \in A^\star$. Formally, for every $a' \neq a \in A$, the half space $\mathcal{H}_{a,a'}$ is defined as:

$$\mathcal{H}_{a,a'} := \left\{ p \in \mathbb{R}^d \mid \langle p, F_a - F_{a'} \rangle \geq c_a - c_{a'} \right\}.$$

Intuitively, $\mathcal{H}_{a,a'}$ is the set of contracts in which the agent prefers $a$ over $a'$. We let $H_{a,a'} := \partial\mathcal{H}_{a,a'}$ be the hyperplane identifying the half space $\mathcal{H}_{a,a'}$.[4] Moreover, we let

---

[4]We assume that, for every $a_1 \neq a_1' \in A$ and $a_2 \neq a_2' \in A$ such that $\{a_1, a_1'\} \neq \{a_2, a_2'\}$, it holds $H_{a_1,a_1'} \neq H_{a_2,a_2'}$.

int($P(a)$) be the interior of the best-response region $P(a)$. Notice that the collection of all the best-response regions $\{P(a)\}_{a \in A^\star}$ constitutes a partition of $\mathcal{P}$.[5]

It is also useful to extend the concept of best-response region to subsets of candidate actions. Given a subset $S \subseteq A$, we let $a(p|S) \in \arg\max_{a \in S} u^A(p, a)$ be the IC action that the agent would select under a contract $p \in \mathbb{R}^d_{\geq 0}$ if the set of available actions were $S$. Moreover, for all $a \in S$, we define the polytope $P(a|S) := \bigcap_{a' \in S} \mathcal{H}_{a,a'} \cap [0,1]^d$, which is the best-response region of action $a$ if $S$ is the set of actions available to the agent. For all $a \in A \setminus S$, we extend the definition of $P(a|S)$ so that $P(a|S) = \varnothing$. Notice that, if $S = A^\star$, then it holds $P(a|S) = P(a)$ for all $a \in A^\star$.

## 3. Toward an Efficient No-Regret Algorithm

We begin by presenting a naïve approach to our problem (Section 3.1). We then introduce the main ideas underpinning our algorithmic approach (Section 3.2).

### 3.1. A Naïve Approach

Next, we describe a naïve approach to tackle our problem, which, however, leads to an algorithm that is neither no-regret nor efficient. This approach is based on the observation that our problem can be reduced to a classic *multi-armed bandit* (MAB) problem, albeit with an exponential number of arms. Specifically, for every possible subset of candidate actions $S \subseteq A$, let $p_S \in \mathcal{P}$ be an optimal contract under the hypothesis that $A^\star = S$. Formally, $p_S \in \arg\max_{p \in \mathcal{P}} u^P(p, a(p|S))$. Then, consider a MAB instance with one arm for each subset $S \subseteq A$. At each round $t \in [T]$, pulling an arm yields as reward a utility value $u^t$ obtained by committing to $p^t = p_S$, where $S \subseteq A$ is the subset associated with the selected arm. Clearly, the mean reward of the arm corresponding to $S \subseteq A$ is $u^P(p_S, a(p_S))$, where we note that $a(p_S)$ is an IC action among the existing ones in $A^\star$. Thus, an optimal arm yields reward OPT, completing the reduction. As a result, a standard algorithm such as UCB1 (Auer et al., 2002) obtains the following result.

**Proposition 3.1.** *There exists a learning algorithm that incurs expected regret $R_T \leq \mathcal{O}(2^K \sqrt{T})$ and whose per-round running time is $\Theta(2^K)$.*

The algorithm in Proposition 3.1 has two major drawbacks: (i) its regret, while sublinear in $T$, scales exponentially with $K$, and (ii) its per-round running time is also exponential in $K$. Both issues stem from the fact that the number of arms in the corresponding MAB instance is exponential in $K$. In the remainder of this paper, we address the two draw-

---

[5]Formally, $\{P(a)\}_{a \in A^\star}$ is a *polyhedral partition* of $\mathcal{P}$, *i.e.*, a cover of $\mathcal{P}$ by polytopes such that int($P(a)$) $\cap$ int($P(a')$) $= \varnothing$ for every $a \neq a' \in A^\star$. In this paper, for brevity, we refer to a polyhedral partition of $\mathcal{P}$ simply as a *partition*.

backs discussed above by designing an algorithm whose regret scales polynomially in $d$ and $K$ and sublinearly in the number of rounds $T$, while also admitting a polynomial per-round running time. As we show in the following, achieving these guarantees comes at the cost of a slightly worse—yet still sublinear—dependence of the regret on $T$.

### 3.2. Our Approach

Next, we introduce the core ideas underlying our learning algorithm, described in detail in Sections 4 and 5.

Our approach originates from the idea of directly learning the set of existing actions $A^\star$. However, discovering all existing actions is intuitively unfeasible, as there may exist two actions whose distributions over outcomes are very similar; distinguishing between them would therefore require an unaffordably large number of rounds. We circumvent this issue by introducing the concept of a *meta-action*, defined as a subset of candidate actions whose distributions over outcomes are similar. Intuitively, the key observation underlying our approach is that the principal's expected utility $u^P(p)$ depends only on the contract $p \in \mathcal{P}$ and on the distribution over outcomes $F_{a(p)}$ of the agent's best response $a(p)$. Therefore, one can focus on meta-actions while incurring only a small loss in the principal's utility. Formally, given any $\epsilon \in (0,1)$, we define an $\epsilon$-*clustering* of the set of candidate actions into meta-actions as follows:

**Definition 3.2.** Given $\epsilon \in (0,1)$, an $\epsilon$-*clustering* $\mathcal{M}^\epsilon \subseteq 2^A$ of $A$ into meta-actions is a partition of the set of candidate actions $A$ such that, for every meta-action $M \in \mathcal{M}^\epsilon$:

(i) for all $a, a' \in M$, it holds $\|F_a - F_{a'}\|_\infty \leq K\epsilon$;
(ii) for all $a \in M, a' \in A \setminus M$, it holds $\|F_a - F_{a'}\|_\infty > \epsilon$.

Any density-based clustering algorithm can be used to efficiently compute an $\epsilon$-clustering (see, *e.g.*, (Han et al., 2011)). Thus, in the description of our learning algorithm in the following sections, we assume access to an $\epsilon$-clustering $\mathcal{M}^\epsilon$, and we let $M^\epsilon(a) \in \mathcal{M}^\epsilon$ be the meta-action containing action $a \in A$. Moreover, we say that a meta-action $M \in \mathcal{M}^\epsilon$ *exists* whenever it contains at least one existing action, *i.e.*, $M \cap A^\star \neq \varnothing$; otherwise, we say that it does *not* exist.

An $\epsilon$-clustering $\mathcal{M}^\epsilon$ induces a polyhedral partition of $\mathcal{P}$ into what we call *meta-best-response regions*.

**Definition 3.3.** Each meta-action $M \in \mathcal{M}^\epsilon$ induces a *meta-best-response region* $P(M) := \bigcup_{a \in M \cap A^\star} P(a)$.

Intuitively, $P(M)$ is the union of the best-response regions of all the existing actions in $M$. Notice that $P(M)$ need *not* be a polytope in general. Moreover, $P(M) \neq \varnothing$ only for the existing meta-actions $M \in \mathcal{M}^\epsilon$. The crucial observation exploited by our algorithm is that knowing an approximation of the meta-best-response regions of existing meta-actions is sufficient to compute an approximately optimal contract.

The approach underlying our learning algorithm can be conceptually split into two sequential phases:

- In the *first* phase (Section 4.2), the algorithm aims to learn existing meta-actions, in order to build a collection of polytopes $\{\widetilde{P}(M)\}_{M \in \mathcal{M}^\epsilon}$ that define a suitable partition of $\mathcal{P}$. Intuitively, each $\widetilde{P}(M)$ should be an approximation of the meta-best-response region $P(M)$.
- In the *second* phase (Section 5), the algorithm exploits the information collected in the first phase. Intuitively, the approximate meta-best-response regions identified earlier can be used to compute an approximately optimal contract, which the algorithm then employs for all remaining rounds while incurring only a small regret.

## 4. Learning Meta-Best-Response Regions

In this section, we describe the first phase of our learning algorithm, which is devoted to learning a suitable partition of $\mathcal{P}$ into approximate meta-best-response regions. Formally, we introduce the following concept of $\epsilon$-*partition*.

**Definition 4.1.** Given any $\epsilon \in (0, 1)$, we say that a partition $\{\widetilde{P}(M)\}_{M \in \mathcal{M}^\epsilon}$ of $\mathcal{P}$ is an $\epsilon$-*partition* whenever, for every meta-action $M \in \mathcal{M}^\epsilon : \widetilde{P}(M) \neq \varnothing$, there is an existing action $a_M \in M \cap A^\star$ such that the following holds:

$$u^{\mathrm{A}}(p, a_M) \geq u^{\mathrm{A}}(p, a(p)) - 2Kd\epsilon \quad \forall p \in \widetilde{P}(M).$$

Intuitively, in an $\epsilon$-partition, each region $\widetilde{P}(M) \neq \varnothing$ constitutes an approximation of its corresponding meta-best-response region $P(M)$, since it is possible to identify an existing action $a_M \in M \cap A^\star$ that is approximately IC under every contract in $\widetilde{P}(M)$. Notice that the agent's best response $a(p)$ under $p \in \widetilde{P}(M)$ may *not* belong to $M$. Nevertheless, there exists an action in $M$ that is approximately IC. This is sufficient to compute an approximately optimal contract. Our algorithm to learn an $\epsilon$-partition has four components:

- A sub-procedure (Query) that "queries" a given contract $p \in \mathcal{P}$ to determine the meta-action $M^\epsilon(a(p))$. The procedure interacts with the agent by committing to $p$ for multiple consecutive rounds (Section 4.1).
- A main procedure (FindPartition) that iteratively refines a partition to find an $\epsilon$-partition (Section 4.2).
- A sub-procedure (TestPartition) that verifies whether a partition is a valid $\epsilon$-partition or not, and, in the latter case, returns a newly discovered existing meta-action that is used by the main procedure to further refine the partition (Section 4.3).
- A sub-procedure (FindPair) that finds a suitable exisiting action that can act as "proxy" for a newly-discovered meta-action (Section 4.4).

The remainder of this section describes these procedures.

### 4.1. Querying Contracts

Algorithm 1 provides the pseudocode for the Query procedure, which intuitively takes as input a contract $p \in \mathcal{P}$ and determines the meta-action to which $a(p)$ belongs. The algorithm leverages the fact that different meta-actions induce sufficiently distinct probability distributions over outcomes and can therefore be distinguished by committing to the contract for a sufficiently small number of rounds.

---

**Algorithm 1** Query

---

**Require:** $p \in \mathcal{P}, \epsilon \in (0, 1), \rho \in (0, 1)$

1: $T_{\epsilon,\rho} \leftarrow \left\lceil \frac{1}{4\epsilon^2} \log \frac{2d}{\rho} \right\rceil$
2: Commit to $p^t = p$ for $T_{\epsilon,\rho}$ rounds
3: Compute $\widehat{F} \in \Delta(\Omega)$ as an empirical estimate of $F_{a(p)}$
4: $a \leftarrow \arg\min_{a' \in A} \|F_{a'} - \widehat{F}\|_\infty$
5: **return** $M^\epsilon(a)$

---

Algorithm 1 commits to $p$ for $T_{\epsilon,\rho} = \Theta(1/\epsilon^2)$ rounds. In this way, it can compute an empirical estimate $\widehat{F}$ of the distribution $F_{a(p)}$ associated with the agent's best response $a(p)$. With high probability, this estimate is at distance at most $\epsilon/2$ from $F_{a(p)}$. Thus, any action $a \in \arg\min_{a' \in A} \|F_{a'} - \widehat{F}\|_\infty$ satisfies $\|F_a - F_{a(p)}\|_\infty \leq \epsilon$. As a result, $a$ and $a(p)$ must belong to the same meta-action, returned by the algorithm. The guarantees provided by Algorithm 1 are the following.

**Lemma 4.2.** *Given $p \in \mathcal{P}$ and $\epsilon, \rho \in (0, 1)$ as input, Algorithm 1 uses less than $\frac{1}{\epsilon^2} \log \frac{2d}{\rho}$ rounds and, with probability at least $1 - \rho$, it returns the meta-action $M^\epsilon(a(p))$.*

### 4.2. Finding a Partition

Algorithm 2 shows the pseudo-code of FindPartition, the main procedure executed by the first phase of our algorithm. The algorithm takes $\epsilon, \rho \in (0, 1)$ and outputs an $\epsilon$-partition with probability at least $1 - (K^2 + K^3/2)\rho$. For ease of presentation, we make the simplifying assumption that the algorithm has access to an existing meta-action $M_{\mathrm{I}} \in \mathcal{M}^\epsilon$, together with one action $a_{M_{\mathrm{I}}} \in M_1 \cap A^\star$ and a contract $p_{M_{\mathrm{I}}} \in \mathrm{int}(P(a_{M_{\mathrm{I}}}))$. In Appendix B.4, we show how this assumption can be easily dropped.

Algorithm 2 works by keeping track of a set of discovered *existing* meta-actions $\mathcal{M}_{\mathrm{E}}^\epsilon \subseteq \mathcal{M}^\epsilon$. Such a set is employed to build an $\epsilon$-partition. To do this, the algorithm also needs to compute, for every meta-action $M \in \mathcal{M}_{\mathrm{E}}^\epsilon$, an *existing* action $a_M \in M \cap A^\star$. Intuitively, the existing action $a_M$ should serve as a "proxy" for the meta-action $M$. Indeed, as formally shown in the following, all existing actions in $M$ have similar distributions over outcomes and similar costs. Thus, the algorithm can approximately treat $a_M$ as the sole action in $M$. By exploiting this observation, Algorithm 2

builds an $\epsilon$-partition $\{\widetilde{P}(M)\}_{M \in \mathcal{M}^\epsilon}$ of $\mathcal{P}$ as:

$$\widetilde{P}(M) := \begin{cases} P(a_M|S) & \text{if } M \in \mathcal{M}^\epsilon_{\mathrm{E}}, \\ \varnothing & \text{if } M \in \mathcal{M}^\epsilon \setminus \mathcal{M}^\epsilon_{\mathrm{E}}, \end{cases} \quad (1)$$

where $S := \{a_M\}_{M \in \mathcal{M}^\epsilon_{\mathrm{E}}}$ is the set of all the existing actions discovered by the algorithm. Intuitively, $\widetilde{P}(M)$ is the best-response region of $a_M$, when the existing actions discovered by the algorithm are the only actions available to the agent.

First, we show that, for every meta-action $M \in \mathcal{M}^\epsilon$, any existing action in $M$ can serve as a "proxy" for $M$.

**Lemma 4.3.** *For any pair of existing actions $a, a' \in A^\star$ belonging to the same meta-action $M \in \mathcal{M}^\epsilon$, it holds:*

$$\|F_a - F_{a'}\|_\infty \leq K\epsilon \ \wedge \ |c_a - c_{a'}| \leq Kd\epsilon.$$

Intuitively, Definition 3.2 already imposes that all actions in $M$ share similar distributions. Furthermore, two actions $a, a' \in M \cap A^\star$ cannot have substantially different costs. Specifically, if $|c_a - c_{a'}| \geq Kd\epsilon$, then one of the two actions would dominate the other, as it would essentially induce the same distribution over outcomes at a lower cost. Thus, since by assumption every existing action is IC for at least one contract, all actions in $M \cap A^\star$ must have similar costs. As a consequence of Lemma 4.3, finding *any* existing action $a_M \in M \cap A^\star$ suffices for Algorithm 2 to work.

Next, we show that there is always a set of existing meta-actions $\mathcal{M}^\epsilon_{\mathrm{E}} \subseteq \mathcal{M}^\epsilon$, with their corresponding "proxy" actions $a_M \in M \cap A^\star$, such that Equation (1) defines an $\epsilon$-partition. Specifically, such a set $\mathcal{M}^\epsilon_{\mathrm{E}}$ consists of all the existing meta-actions, so that $M \cap A^\star = \varnothing$ for all $M \notin \mathcal{M}^\epsilon_{\mathrm{E}}$. First, we observe that, any existing action in $M \cap A^\star$ for a given $M \in \mathcal{M}^\epsilon$ yields essentially the same agent's utility.

**Lemma 4.4.** *For every $M \in \mathcal{M}^\epsilon$ and pair of existing actions $a, a' \in \mathcal{M}^\epsilon$, the following holds for every $p \in \mathcal{P}$:*

$$u^A(p, a') - 2Kd\epsilon \leq u^A(p, a) \leq u^A(p, a') + 2Kd\epsilon.$$

Now, suppose that the set $\mathcal{M}^\epsilon_{\mathrm{E}}$ computed by Algorithm 2 consists of all the existing meta-actions. For any $M \in \mathcal{M}^\epsilon_{\mathrm{E}}$ and $p \in \widetilde{P}(M)$, let $M' = M^\epsilon(a(p))$ be the meta-action of the agent's best response $a(p)$. Then, we have that:

$$u^A(p, a_M) \geq u^P(p, a_{M'}) \geq u^P(p, a(p)) - 2Kd\epsilon,$$

which holds by Equation (1) and Lemma 4.4. Therefore, $a_M$ is approximately IC under all the contracts in $\widetilde{P}(M)$, and, thus, Equation (1) defines an $\epsilon$-partition as desired.

We are now ready to describe how Algorithm 2 works. As a first step, the algorithm initializes the set $\mathcal{M}^\epsilon_{\mathrm{E}}$ to $M_1$, and the set of "proxy" actions $S$ to the single action $a_{M_1}$ (recall that suitable $M_1$ and $a_{M_1}$ are assumed to be available throughout this section, for ease of presentation). The

---

**Algorithm 2** FindPartition

**Require:** $\epsilon \in (0, 1)$, $\rho \in (0, 1)$
    ▷ Assume access to an existing meta-action $M_{\mathrm{I}} \in \mathcal{M}^\epsilon$, together with $a_{M_{\mathrm{I}}} \in M_{\mathrm{I}} \cap A^\star$ and $p_{M_{\mathrm{I}}} \in \mathrm{int}(P(a_{M_{\mathrm{I}}}))$
1: $\mathcal{M}^\epsilon_{\mathrm{E}} \leftarrow \{M_{\mathrm{I}}\}, S \leftarrow \{a_{M_{\mathrm{I}}}\}$
2: **while** TestPartition$(\mathcal{M}^\epsilon_{\mathrm{E}}, S, p_{M_{\mathrm{I}}}, \epsilon, \rho) \neq \bot$ **do**
3:     $a_{\widehat{M}} \leftarrow$ The action returned by TestPartition
4:     $\mathcal{M}^\epsilon_{\mathrm{E}} \leftarrow \mathcal{M}^\epsilon_{\mathrm{E}} \cup \{\widehat{M}\}$
5:     $S \leftarrow S \cup \{a_{\widehat{M}}\}$
6: **end while**
7: **return** $\{\widetilde{P}(M)\}_{M \in \mathcal{M}^\epsilon}$ built according to Equation (1)

---

algorithm then proceeds iteratively, with the goal of expanding the sets $\mathcal{M}^\epsilon_{\mathrm{E}}$ and $S$ until Equation (1) defines an $\epsilon$-partition. At the beginning of each iteration, it verifies whether Equation (1) defines an $\epsilon$-partition or not. This task is delegated to the TestPartition sub-procedure, described in detail in Section 4.3. When the test is successful (*i.e.*, TestPartition returns $\bot$), Algorithm 2 computes an $\epsilon$-partition according to Equation (1) and terminates. Otherwise, when the test is unsuccessful, TestPartition provides an action $a_{\widehat{M}} \in \widehat{M} \cap A^\star$ for some $\widehat{M} \notin \mathcal{M}^\epsilon_{\mathrm{E}}$. Therefore, the algorithm adds $\widehat{M}$ to $\mathcal{M}^\epsilon_{\mathrm{E}}$ and $a_{\widehat{M}}$ to $S$, so that the next iteration will test a refined partition. Crucially, TestPartition discovers a new existing meta-action whenever unsuccessful, so that the partition can be refined.

Since Equation (1) defines an $\epsilon$-partition when all the existing meta-actions are known and $|\mathcal{M}^\epsilon| \leq K$, Algorithm 2 terminates in at most $K$ iterations. It may terminate earlier, as only a subset of existing meta-actions may be enough to build an $\epsilon$-partition. By accounting for the number of rounds required by TestPartition (see Lemma 4.7):

**Lemma 4.5.** *Given any $\epsilon \in (0, 1)$ and $\rho \in (0, 1)$ as input, Algorithm 2 computes an $\epsilon$-partition of $\mathcal{P}$ in $\frac{K^3}{\epsilon^2} \log \frac{2d}{\rho}$ rounds, with probability at least $1 - (2K^2 + K^3/2)\rho$.*

### 4.3. Testing a Partition

TestPartition (Algorithm 3) is the sub-procedure employed by Algorithm 2 to determine whether the partition defined by Equation (1) is an $\epsilon$-partition or *not*. Specifically, a correct execution of Algorithm 3 is defined as follows.

**Definition 4.6.** Algorithm 3 correctly terminates its execution in one of two possible cases:

(i) when it returns $a_{\widehat{M}} \in \widehat{M} \cap A^\star$ and $p_{\widehat{M}} \in \mathrm{int}(P(a_{\widehat{M}}))$ for some existing meta-action $\widehat{M} \in \mathcal{M}^\epsilon \setminus \mathcal{M}^\epsilon_{\mathrm{E}}$; or

(ii) when it returns $\bot$ and Equation (1) is an $\epsilon$-partition.

Algorithm 3 correctly terminates when either identifies (i) a new existing meta-action or (ii) concludes that the current partition is an $\epsilon$-partition. In case (i), it also returns an

---

**Algorithm 3** TestPartition

**Require:** $\mathcal{M}_\text{E}^\epsilon \subseteq \mathcal{M}^\epsilon$, A set $S := \{a_M\}_{M \in \mathcal{M}_\text{E}^\epsilon}$ of actions, A contract $p_{M_\text{I}} \in \text{int}(P(M_\text{I}))$ for some $M_\text{I} \in \mathcal{M}_\text{E}^\epsilon$, $\epsilon \in (0,1), \rho \in (0,1)$

1: **for all** $a \in A \setminus \bigcup_{M \in \mathcal{M}_\text{E}^\epsilon} M$ **do**
2: $\quad$ $C(a) \leftarrow \{p \in \mathcal{P} \mid p \text{ satisfies Equation (2)}\}$
3: $\quad$ **if** $\text{vol}(C(a)) \neq 0$ **then**
4: $\quad\quad$ Sample $p$ uniformly at random from $C(a)$
5: $\quad\quad$ $\widehat{M} \leftarrow \text{Query}(p, \epsilon, \rho)$
6: $\quad\quad$ **if** $\widehat{M} \notin \mathcal{M}_\text{E}^\epsilon$ **then**
7: $\quad\quad\quad$ $(a_{\widehat{M}}, p_{\widehat{M}}) \leftarrow \text{FindPair}(\widehat{M}, p, M_\text{I}, p_{M_\text{I}}, \epsilon, \rho)$
8: $\quad\quad\quad$ **return** $(a_{\widehat{M}}, p_{\widehat{M}})$
9: $\quad\quad$ **end if**
10: $\quad$ **end if**
11: **end for**
12: **return** $\perp$

---

existing action $a_{\widehat{M}} \in \widehat{M} \cap A^\star$ for the newly-discovered meta-action and a contract $p_{\widehat{M}}$ in the interior of its best-response region. While $p_{\widehat{M}}$ is *not* employed by Algorithm 2, it is crucial for developing the procedure in Appendix B.4.

Algorithm 3 iterates over the actions in $A$ that do *not* belong to any existing meta-action in $\mathcal{M}_\text{E}^\epsilon$, looking for an existing one that "breaks" the partition. An action $a \in A \setminus A_\text{E}$, with $A_\text{E} := \bigcup_{M \in \mathcal{M}_\text{E}^\epsilon} M$, is said to "break" the partition when:

$$u^\text{A}(p, a) \geq u^\text{A}(p, a_M) + 2Kd\epsilon \quad \forall M \in \mathcal{M}_\text{E}^\epsilon \quad (2)$$

for some contract $p \in \mathcal{P}$ under which $a$ is IC. Intuitively, there is no existing action $a_M \in S$ approximately IC under such a contract and, thus, Equation (1) cannot define an $\epsilon$-partiton. To identify an action that "breaks" the partition, for each $a \in A \setminus A_\text{E}$, Algorithm 3 computes the polytope $\mathcal{C}(a)$ consisting of all the contracts $p \in \mathcal{P}$ satisfying Equation (2). Then, the algorithm looks for $a$ in the interior of this set, *i.e.*, it looks for a contract $p \in \text{int}(\mathcal{C}(a))$ such that $a(p) = a$. If it succeeds, then $a$ "breaks" the partition; otherwise, Equation (1) is an $\epsilon$-partition, and it returns $\perp$. Specifically, for every $a \in A \setminus A_\text{E}$, Algorithm 3 calls Query on a contract $p \in \text{int}(\mathcal{C}(a))$—if the set has empty interior, it skips to the next action (Line 3). Let $\widehat{M} \in \mathcal{M}^\epsilon$ be the found meta-action. Then, two different cases are possible. If $\widehat{M} \in \mathcal{M}_\text{E}^\epsilon$, then $a \notin A^\star$, as the agent prefers $a$ over every existing action $a' \in A_\text{E} \cap A^\star$ under contracts in $\mathcal{C}(a)$. As a result, $a$ does *not* "break" the partition, and the algorithm continues to the next action. Instead, if $\widehat{M} \notin \mathcal{M}_\text{E}^\epsilon$, then $a$ may "break" the partition. In such a case, Algorithm 3 looks for $(a_{\widehat{M}}, p_{\widehat{M}})$ with $a_{\widehat{M}} \in \widehat{M} \cap A^\star$ and $p_{\widehat{M}} \in \text{int}(P(a_{\widehat{M}}))$ for some $\widehat{M} \in \mathcal{M}^\epsilon \setminus \mathcal{M}_\text{E}^\epsilon$. To this end, the algorithm employs FindPair, described in detail in the following section. Subsequently, Algorithm 3 terminates its execution and returns $(a_{\widehat{M}}, p_{\widehat{M}})$.

Finally, Algorithm 3 always employs at most $K$ calls to

---

**Algorithm 4** FindPair

**Require:** Two different meta-actions $M_1 \neq M_2 \in \mathcal{M}^\epsilon$, Two contracts $p_1 \in \text{int}(P(M_1))$ and $p_2 \in \text{int}(P(M_2))$, $\epsilon \in (0,1), \rho \in (0,1)$

1: Let $L$ be the line segment connecting $p_1$ and $p_2$
2: $\mathcal{I} \leftarrow \{p^\circ \in \text{int}(L) \mid p^\circ \in H_{a,a'}, a \in M_1, a' \notin M_1\}$
3: $a_{M_1} \leftarrow \perp$
4: **while** $a_{M_1} = \perp$ **do**
5: $\quad$ $p^\circ \leftarrow$ Any contract in $\mathcal{I}$
6: $\quad$ Let $a \in M_1, a' \notin M_1$ be such that $p^\circ \in H_{a,a'}$
7: $\quad$ $p, p' \leftarrow \text{ComputeClosePoints}(p_1, p_2, a, a')$
8: $\quad$ $M \leftarrow \text{Query}(p, \epsilon, \rho), M' \leftarrow \text{Query}(p', \epsilon, \rho)$
9: $\quad$ **if** $M \neq M'$ **then**
10: $\quad\quad$ $a_{M_1} \leftarrow a, p_{M_1} \leftarrow p$
11: $\quad$ **end if**
12: $\quad$ $\mathcal{I} \leftarrow \mathcal{I} \setminus \{p^\circ\}$
13: **end while**
14: **return** $(a_{M_1}, p_{M_1})$

---

Query, one for each loop iteration. Moreover, when it terminates in case (i), it requires an additional $K^2/2$ calls to Query to run FindPair. Thus, we can prove:

**Lemma 4.7.** *Suppose that Algorithm 3 is called with the following inputs: a nonempty set of existing meta-actions $\mathcal{M}_\text{E}^\epsilon \subseteq \mathcal{M}^\epsilon$; a set $S := \{a_M\}_{M \in \mathcal{M}_\text{E}^\epsilon}$ with $a_M \in M \cap A^\star$ for every $M \in \mathcal{M}_\text{E}^\epsilon$; a contract $p_{M_\text{I}} \in \text{int}(P(M_\text{I}))$ for some $M_\text{I} \in \mathcal{M}_\text{E}^\epsilon$; $\epsilon, \rho \in (0,1)$. Then, Algorithm 3 correctly terminates (according to Definition 4.6) with probability at least $1 - (2K + K^2/2)\rho$, by using $\frac{K^2}{\epsilon^2} \log \frac{2d}{\rho}$ rounds.*

## 4.4. Finding Existing Actions

The sub-procedure FindPair (Algorithm 4) takes: the new meta-action $M_1$, a contract $p_1 \in \text{int}(P(M_1))$, and another (known) existing meta-action $M_2$ with a contract $p_2 \in \text{int}(P(M_2))$. Then, it finds an action $a_{M_1} \in M_1 \cap A^\star$ and a contract $p_{M_1}$ under which $a_{M_1}$ is IC.

Let $L$ be the line segment connecting $p_1$ and $p_2$. This is guaranteed to intersect the boundary of $P(M_1)$, as one of its extremes is in $\text{int}(P(M_2))$. Specifically, it intersects hyperplanes $H_{a,a'}$, with $a \in M_1$ and $a' \notin M_1$, in a finite set of points $\mathcal{I}$; one of these points is on the boundary of $P(M_1)$. Algorithm 4 iterates over these contracts $p^\circ \in \mathcal{I}$. At each iteration, given $p^\circ \in L \cap H_{a,a'}$, it computes two contracts $p, p' \in L$ sufficiently close to $p^\circ$, so as to guarantee that the line segment connecting them intersects the separating hyperplane $H_{a,a'}$ and no others (see Appendix B.3 for details). Therefore, either $a(p) = a$, $a(p') = a'$ or $a(p) = a(p')$. Algorithm 4 then proceeds by calling Query on $p$ and $p'$, finding the meta-actions $M$ and $M'$, respectively. If $p^\circ$ is on the boundary of $P(M_1)$, then $M \neq M'$ and it must be $a(p) = a$ and $a(p') = a'$. As a result, it can return $a_{M_1} = a$

**Algorithm 5** `ComputeContract`

---

**Require:** $\epsilon$-partition $\{\widetilde{P}(M)\}_{M \in \mathcal{M}^\epsilon}$ (see Definition 4.1),
A set $\{a_M\}_{M \in \mathcal{M}^\epsilon}$ with $a_M \in M$ for every $M \in \mathcal{M}^\epsilon$.
 1: $(\widetilde{p}, \widetilde{M}) \leftarrow \arg\max_{M \in \mathcal{M}^\epsilon, \, p \in \widetilde{P}(M)} \langle r - p, F_{a_M} \rangle$
 2: $\widetilde{p}^\diamond \leftarrow \left( 1 - \sqrt{2Kd\epsilon} \right) \widetilde{p} + r\sqrt{2Kd\epsilon}$
 3: **return** $\widetilde{p}^\diamond$

---

and $p_{M_1} = p$. Otherwise, when $p^\circ$ is not on the boundary of $P(M_1)$, $M = M'$ and the algorithm advances to the next iteration. By observing that $|\mathcal{I}| \leq \mathcal{O}(K^2)$, we can prove:

**Lemma 4.8.** *Suppose that Algorithm 4 is called with the following inputs: two different meta-actions $M_1 \neq M_2 \in \mathcal{M}^\epsilon$; two contracts $p_1 \in \text{int}(P(M_1))$ and $p_2 \in \text{int}(P(M_2))$; $\epsilon, \rho \in (0, 1)$. Moreover, assume that the line segment connecting $p_1$ and $p_2$ never intersects multiple separating hyperplanes in a single point. Then, Algorithm 4 computes $a_{M_1} \in A^\star \cap M_1$ and $p_{M_1} \in \text{int}(P(a_{M_1}))$ by using at most $\frac{K^2}{2\epsilon^2} \log \frac{2d}{\rho}$ rounds, with probability at least $1 - \rho K^2/2$.*

Let us observe that Lemma 4.8 assumes that $L$ never intersects multiple separating hyperplanes in a single point $p^\circ$. Since Algorithm 3 samples $p_2$ uniformly at random from a fully-dimensional polytope, this holds almost surely.

## 5. Finding Approximately Optimal Contracts

In this section, we describe how to employ an $\epsilon$-partition to compute an approximately optimal contract to be used in the second phase. Specifically, we provide Algorithm 5, which takes as inputs the $\epsilon$-partition $\{\widetilde{P}(M)\}_{M \in \mathcal{M}^\epsilon}$ learned by Algorithm 2 and an action $a_M \in M$ for each $M \in \mathcal{M}^\epsilon$, and outputs an approximately optimal contract $\widetilde{p}^\diamond$. Let us observe that one can use the same actions $a_M$ for $M \in \mathcal{M}^\epsilon$ computed by Algorithm 2, but this is *not* strictly necessary.

First, for each $M \in \mathcal{M}^\epsilon$, the algorithm computes a utility-maximizing contract in the set $\widetilde{P}(M)$ under the assumption that the agent selects $a_M$. This amounts to solving an LP for each $M \in \mathcal{M}^\epsilon$, maximizing $u^P(p, a_M)$ over $p \in \widetilde{P}(M)$. We denote as $\widetilde{p}$ the best contract among those computed in this way, and we let $\widetilde{M}$ be its meta-action. Formally:

$$(\widetilde{p}, \widetilde{M}) \in \arg\max_{M \in \mathcal{M}^\epsilon, \, p \in \widetilde{P}(M)} \langle r - p, F_{a_M} \rangle.$$

We show that $u^P(\widetilde{p}, a_{\widetilde{M}})$ is close to the value of OPT:

**Lemma 5.1.** *Suppose that Algorithm 5 is called with inputs an $\epsilon$-partition $\{\widetilde{P}(M)\}_{M \in \mathcal{M}^\epsilon}$ and a set $\{a_M\}_{M \in \mathcal{M}^\epsilon}$ with $a_M \in M$ for every $M \in \mathcal{M}^\epsilon$. Then:*

$$u^P(\widetilde{p}, a_{\widetilde{M}}) \geq \text{OPT} - Kd\epsilon - 2\sqrt{2Kd\epsilon}.$$

Notice that the lemma does *not* provide a lower bound on the actual expected utility $u^P(\widetilde{p})$, but rather on $u^P(\widetilde{p}, a_{\widetilde{M}})$.

However, the agent's best response $a(p)$ might not even belong to $\widetilde{M}$, as the region $\widetilde{P}(\widetilde{M})$ does not coincide with the true region $P(\widetilde{M})$. To circumvent this issue, Algorithm 5 computes $\widetilde{p}^\diamond$ as an opportune linear combination between $\widetilde{p}$ and the reward vector $r$. This step, developed by Dutting et al. (2021), guarantees that $u^P(\widetilde{p}^\diamond) \geq u^P(\widetilde{p}, a_{\widetilde{M}}) - \mathcal{O}(\sqrt{\epsilon})$.

**Lemma 5.2.** *Suppose that Algorithm 5 is called with inputs an $\epsilon$-partition $\{\widetilde{P}(M)\}_{M \in \mathcal{M}^\epsilon}$ and a set $\{a_M\}_{M \in \mathcal{M}^\epsilon}$ with $a_M \in M$ for every $M \in \mathcal{M}^\epsilon$. Then:*

$$u^P(\widetilde{p}^\diamond) \geq \text{OPT} - (4\sqrt{2} + 2)Kd\sqrt{\epsilon}.$$

## 6. Putting Everything Together

We are finally ready to design our two-phase learning algorithm and prove its regret guarantees. To do so, we assemble all the algorithms described in Sections 4 and 5. Specifically, our algorithm attains regret sublinear in $T$ and polynomial in the instance size, characterized by $K$ and $d$, with probability at least $1 - \delta$, where $\delta \in (0, 1)$ is an input parameter.

Let $\epsilon = \Theta((K^2/dT)^{2/5})$ and $\rho = \Theta(\delta/K^3)$. First, our algorithm finds an initial existing meta-action $M_I \in \mathcal{M}^\epsilon$, together with an existing action $a_{M_I} \in M \cap A^\star$ and a contract $p_{M_I} \in \text{int}(P(a_{M_I}))$—see Appendix B.4. Then, in the *first* phase, the algorithm executes the `FindPartition` procedure to find a suitable $\epsilon$-partition of $\mathcal{P}$. This takes $T_E \leq \frac{2K^3}{\epsilon^2} \log \frac{2d}{\rho}$ rounds. If the available number of rounds $T$ is not large enough, *i.e.*, $T \leq T_E$, the algorithm immediately stops as soon as $T$ rounds have been completed.

In the *second* phase, the algorithm computes an approximately optimal contract $p^\circ$ by means of Algorithm 5. Then, it commits to $p^t = p^\circ$ for all the remaining rounds $t$, incurring a per-round regret of $\mathcal{O}(Kd\sqrt{\epsilon})$. The full pseudo-code of our learning algorithm is provided in Appendix A. The following theorem states its no-regret guarantee.

**Theorem 6.1.** *There exists a learning algorithm that, given any $\delta \in (0, 1)$ and $T \in \mathbb{N}$ as inputs, incurs in a regret*

$$R_T \leq \mathcal{O}\left( K^{\frac{7}{5}} d^{\frac{4}{5}} T^{\frac{4}{5}} \log\left( \frac{dK^3}{\delta} \right) \right)$$

*with probability at least $1 - \delta$.*

Let us remark that our algorithm is also computationally efficient. Moreover, notice that its regret guarantees improve upon existing approaches when $K \leq \mathcal{O}(\text{poly}(n))$, since in this regime its regret scales polynomially in both $n = |A^\star|$ and $d$. By contrast, when the agent's technology is small, *e.g.*, $n = \mathcal{O}(\log(K))$, the algorithm incurs in a regret comparable to that achieved by Bacchiocchi et al. (2024), *i.e.*, exponential in $n$. However, when the agent's technology is small, an exponential dependence on its size is acceptable.

## 7. Conclusions

Prior knowledge of a superset $A$ of the true agent's technology $A^\star$ allows one to design an efficient and no-regret algorithm, whose regret is polynomial in the instance size and sublinear in the time horizon. Let us remark that an instance in this setting includes $K > n = |A^\star|$ actions, *i.e.*, the entire superset of actions provided as input to the algorithm. Hence, the regret scales polynomially in $n$ whenever the size $K$ of the set of candidate actions itself is polynomial in $n$; otherwise, the performance of Algorithm 6 can be comparable to that of (Bacchiocchi et al., 2024).

We leave open the problem of deriving a lower bound on the regret in this setting. While Algorithm 6 attains $\widetilde{\mathcal{O}}(\text{poly}(K,d)T^{4/5})$ regret, a naïve and inefficient approach suffers $\mathcal{O}(2^K\sqrt{T})$ regret. Whether removing the exponential dependence on $K$ necessarily worsens the dependence on $T$ remains an open question.

Finally, let us observe that when the principal does *not* know a superset of possibly available actions, Zhu et al. (2023) provide a lower bound of $\otimes(T^{\frac{d+1}{d+2}})$ when there are $n = \Theta(T^{\frac{d}{d+2}}) > T^{1/2}$ actions. However, no lower bound is known for a generic number of actions $n > 0$.

## Acknowledgements

MB and MC are funded by the EU Horizon project ELIAS (European Lighthouse of AI for Sustainability, No. 101120237). AM is funded by Ministero dell'Università e della ricerca pursuant to D.D. n. 18010 of 12 November 2025 – BANDO FIS 3, project FIS-2024-05736 (Starting Grant), title: "Towards a trustworthy strategic use of data in machine learning pipelines " (STRATDATA), CUP: D53C25002380001.

## Impact Statement

This paper presents work whose goal is to advance the field of Machine Learning. There are many potential societal consequences of our work, none of which we feel must be specifically highlighted here.

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

## A. Full Algorithm and Proof of Theorem 6.1

---
**Algorithm 6** `MainAlgorithm`

---
**Require:** $T \in \mathbb{N}, \delta \in (0, 1)$
1:  $\epsilon \leftarrow \left(\frac{K^2}{dT}\right)^{2/5}$
2:  $\rho \leftarrow \frac{\delta}{5K^3}$
3:  execute `FindInitialPoint`$(\epsilon, \rho)$                  ▷ Can return a pair $(a_{M_1}, p_{M_1})$ or an $\epsilon$-partition
4:  **if** `FindInitialPoint` returned $a_{M_1}, p_{M_1}$ **then**
5:      $\mathcal{Y} \leftarrow$ `FindPartition`$(a_{M_1}, p_{M_1}, \epsilon, \rho)$
6:  **else**
7:      $\mathcal{Y} \leftarrow$ the partition returned by `FindInitialPoint`
8:  **end if**
9:  $p^\circ \leftarrow$ `Commit`$(\mathcal{Y}, \epsilon)$
10: Commit to $p^\circ$ for the remaining rounds

---

**Theorem 6.1.** *There exists a learning algorithm that, given any $\delta \in (0, 1)$ and $T \in \mathbb{N}$ as inputs, incurs in a regret*

$$R_T \leq \mathcal{O}\left(K^{\frac{7}{5}} d^{\frac{4}{5}} T^{\frac{4}{5}} \log\left(\frac{dK^3}{\delta}\right)\right)$$

*with probability at least $1 - \delta$.*

*Proof.* Suppose that $T$ is large enough, namely $T \geq T_{\mathrm{E}}$ for a quantity $T_{\mathrm{E}}$ that we will define later. We show that in this case, with probability at least $1 - \delta$ the algorithm executes Line 10 (i.e., it completes the exploration phase) and incurs a regret at most:

$$R_T \leq \mathcal{O}\left(K^{\frac{7}{5}} d^{\frac{4}{5}} T^{\frac{4}{5}} \log\left(\frac{dK^3}{\delta}\right)\right).$$

Let $\epsilon = \left(\frac{K^2}{dT}\right)^{2/5}$ and $\rho = \frac{\delta}{5K^3}$ as defined by Algorithm 6. At Line 3, the algorithm invokes the procedure `FindInitialPoint`. With probability at least $1 - (3K + 2K^2 + K^3/2)\rho$, this procedure terminates in at most:

$$T_{\mathrm{I}} := \frac{2K^2}{\epsilon^2} \log\left(\frac{2d}{\rho}\right)$$

rounds, and it computes either an $\epsilon$-cover or a pair $(a_{M_1}, p_{M_1})$ suitable for `FindPartition` (Lemma B.4). In the latter case, Algorithm 6 invokes `FindPartition`. By Lemma 4.5, this procedure computes an $\epsilon$-partition in a number of rounds less than:

$$T_{\mathrm{F}} := \frac{K^3}{\epsilon^2} \log\left(\frac{2d}{\rho}\right),$$

with probability at least $1 - (2K^2 + K^3/2)\rho$. Overall, with probability at least $1 - (3K + 4K^2 + K^3)\rho$, Algorithm 6 computes an $\epsilon$-partition in at most $T_{\mathrm{I}} + T_{\mathrm{F}}$. We let $T_{\mathrm{E}} \geq T_{\mathrm{I}} + T_{\mathrm{F}}$ be an upper bound of this number defined as:

$$T_{\mathrm{E}} := \frac{2K^3}{\epsilon^2} \log\left(\frac{2d}{\rho}\right).$$

Observe that by taking $\rho \leq 5K^{-3}$, this event has probability at least $1 - \delta$. We will continue the analysis conditioned on such an event.

According to Lemma 5.2, the contract $p^\circ$ Algorithm 6 commits to provides utility $u^{\mathrm{P}}(p^\circ) \geq \mathrm{OPT} - (4\sqrt{2} + 2)Kd\sqrt{\epsilon}$. Hence, accounting for a per-round regret of at most 2 during the initial $T_{\mathrm{E}}$ rounds, and of $\mathcal{O}(Kd\sqrt{\epsilon})$ during the subsequent rounds, we can bound the regret of Algorithm 6 as:

$$R_T \leq \mathcal{O}\left(\frac{K^3}{\epsilon^2} \log\left(\frac{d}{\rho}\right) + TKd\sqrt{\epsilon}\right)$$

$$= \mathcal{O}\left(K^{\frac{7}{5}} d^{\frac{4}{5}} T^{\frac{4}{5}} \log\left(\frac{dK^3}{\delta}\right)\right),$$

where the last equality is obtained substituting the values of $\epsilon = \left(\frac{K^2}{dT}\right)^{2/5}$ and $\rho = \frac{\delta}{5K^3}$.

We conclude the proof by noticing that the upper bound still holds even when $T < T_{\mathrm{E}}$, as in such a case our upper bound is larger than the time horizon. $\qquad\square$

## B. Proofs and Algorithms omitted from Section 4

**Lemma 4.3.** *For any pair of existing actions $a, a' \in A^\star$ belonging to the same meta-action $M \in \mathcal{M}^\epsilon$, it holds:*

$$\|F_a - F_{a'}\|_\infty \le K\epsilon \ \wedge \ |c_a - c_{a'}| \le Kd\epsilon.$$

*Proof.* Let $\bar{p}$ be a contract such that $a(\bar{p}) = a$, which exists as $a \in A^\star$. By the definition of $a(\bar{p})$, we have:

$$\langle F_a, \bar{p}\rangle - c_a \ge \langle F_{a'}, \bar{p}\rangle - c_{a'}$$

for every $a' \in A^\star$. Rearranging we get:

$$(c_a - c_{a'}) \le \langle F_a - F_{a'}, \bar{p}\rangle$$

Since both $a$ and $a'$ belong to the $\epsilon$-meta-action $M$, we also have that $\|F_a - F_{a'}\| \le K\epsilon$. Hence $\langle F_a - F_{a'}, \bar{p}\rangle \le Kd\epsilon$, which implies that $(c_a - c_{a'}) \le Kd\epsilon$. Rearranging we get:

$$c_a \le c_{a'} + Kd\epsilon. \tag{3}$$

At the same time, for every $a' \in A^\star$, there exists some contract $\bar{p}'$ such that $a(\bar{p}') = a'$. By the same argument provided above, it is possible to show that $c_{a'} \le c_a + Kd\epsilon$. As a result, we have $|c_a - c_{a'}| \le Kd\epsilon$. Since the two actions belong to the same meta-action, we also have $\|F_a - F_{a'}\|_\infty \le K\epsilon$, concluding the proof. $\qquad\square$

**Lemma 4.4.** *For every $M \in \mathcal{M}^\epsilon$ and pair of existing actions $a, a' \in \mathcal{M}^\epsilon$, the following holds for every $p \in \mathcal{P}$:*

$$u^A(p, a') - 2Kd\epsilon \le u^A(p, a) \le u^A(p, a') + 2Kd\epsilon.$$

*Proof.* Consider any $p \in \mathcal{P}$. We have:

$$\begin{aligned}
|u^{\mathrm{A}}(p, a_M) - u^{\mathrm{A}}(p, a)| &= |\langle F_{a_M}, p\rangle - c_{a_M} - \langle F_a, p\rangle + c_a| \\
&= |\langle F_{a_M} - F_a, p\rangle + (c_a - c_{a_M})| \\
&\le |\langle F_{a_M} - F_a, p\rangle| + |c_a - c_{a_M}| \\
&\le Kd\epsilon + Kd\epsilon = 2Kd\epsilon,
\end{aligned}$$

where the first inequality is the triangle inequality, and the second holds as $|\langle F_{a_M} - F_a, p\rangle| \le Kd\epsilon$ since both actions belong to $M$, and $|c_a - c_{a_M}| \le Kd\epsilon$ by Lemma 4.3. $\qquad\square$

**Lemma 4.5.** *Given any $\epsilon \in (0, 1)$ and $\rho \in (0, 1)$ as input, Algorithm 2 computes an $\epsilon$-partition of $\mathcal{P}$ in $\frac{K^3}{\epsilon^2} \log \frac{2d}{\rho}$ rounds, with probability at least $1 - (2K^2 + K^3/2)\rho$.*

*Proof.* We let $\mathcal{M}_{\mathrm{E}}^{\epsilon,i}$ be the set of meta-action computed by Algorithm 2 before the $i$-th execution of Algorithm 3, and $S^i := \{a_M^i\}_{M \in \mathcal{M}_{\mathrm{E}}^{\epsilon,i}}$ the corresponding set of actions.

For the purpose of this proof, we say that the $i$-th iteration has been correctly executed when it takes than:

$$N := \frac{K^2}{\epsilon^2} \log\left(\frac{2d}{\rho}\right)$$

rounds, and the test has been executed correctly according to Definition 4.6.

Consider any iteration $i$ of the algorithm. Under the event that all previous iteration were executed correctly, all these previous tests returned new representative actions, as otherwise the algorithm would have stopped before the $i$-th iteration. Consequently, $|\mathcal{M}_E^{\epsilon,i}| = i$, as initially $|\mathcal{M}_E^{\epsilon,1}| = 1$ and each of the previous $(i-1)$ iterations added a new action. Furthermore, the set $S^i$ consists of actions $a_M \in M \cap A^\star$, $M \in \mathcal{M}_E^{\epsilon,i}$. By Lemma 4.7, we have that the $i$-th iteration is executed correctly with probability at least $1 - (2K + K^2/2)\rho$. As there may exists up to $K$ different meta-actions, it must be the case that $i \le K$. Therefore, the algorithm can execute at most $K$ successive iterations correctly, after which Equation (1) defines an $\epsilon$-partition (a correctly executed test must return $\perp$ when $\mathcal{M}_E = \mathcal{M}$, as there are no other meta-actions to discover). Employing an union bound over these iterations, we have that with probability at least $1 - (2K^2 + K^3/2)\rho$, Algorithm 2 returns an $\epsilon$-partition in at most $KN$ rounds, concluding the proof. $\qquad\square$

## B.1. Query

**Lemma 4.2.** *Given $p \in \mathcal{P}$ and $\epsilon, \rho \in (0,1)$ as input, Algorithm 1 uses less than $\frac{1}{\epsilon^2} \log \frac{2d}{\rho}$ rounds and, with probability at least $1 - \rho$, it returns the meta-action $M^\epsilon(a(p))$.*

*Proof.* Suppose that the estimator computed by Algorithm 1 satisfy:

$$\|F_{a(p)} - \widehat{F}\|_\infty \le \frac{\epsilon}{2} \qquad (4)$$

Let $a^\circ \in \arg\min_{a \in A} \|F_a - \widehat{F}\|_\infty$ be the action selected by Algorithm 1 at Line 4. We have:

$$\|F_a - \widehat{F}_\omega\|_\infty \le \|F_{a(p)} - \widehat{F}\|_\infty \le \frac{\epsilon}{2},$$

where the first inequality holds due to the definition of $a$ and the second thanks to Equation 4. Therefore $\|F_a - F_{a(p)}\|_\infty \le \epsilon$, and the both $a$ and $a(p)$ belong to the same $\epsilon$-meta-action. As a result, Algorithm 1 returns the correct meta-action when Equation 4 holds.

In order to conclude the proof, let $T_{\epsilon,\rho}$ the number rounds employed by Algorithm 1. By an union bound over the $d$ outcomes and Hoeffding inequality, Equation 4 holds with probability at least $1 - 2d \exp(-2T_{\epsilon,\rho}\epsilon^2)$. As Algorithm 1 employs:

$$\frac{1}{4\epsilon^2} \log(2d/\rho) \le T_{\epsilon,\rho} \le \frac{1}{\epsilon^2} \log(2d/\rho)$$

rounds, we have that Equation 4 holds with probability at least $1 - \rho$. $\qquad\square$

## B.2. Test Cover

We now prove Lemma 4.7, which states the main guarantees of Algorithm 3: that it terminates correctly (according to Definition 4.6) in a given number of rounds with high probability. We recall that Algorithm 3 can either return a new pair $(a_{\widehat{M}}, p_{\widehat{M}})$ or $\perp$. Hence, we need to show that, with high probability, (1) it terminates in $\frac{K^2}{2\epsilon^2} \log(2d/\rho)$, (2) if it returns a pair $(a_{\widehat{M}}, p_{\widehat{M}})$, then such a pair correctly represents a new meta-action, and (3) if it returns $\perp$, then an $\epsilon$-partition has been found.

We decompose the proof of Lemma 4.7 into two different lemmas. With the first lemma (Lemma B.1), we prove (1) and (2). Crucially, Lemma B.1 does not assume in input existing meta-actions $a_M \in M \cap A^\star$ for every $M$. This allows us to use Algorithm 3 in Section B.3 with different guesses for the existing actions and still bound the number of rounds when the guess is wrong.

The second lemma (Lemma B.2) proves instead that, given existing actions $a_M \in M \cap A^\star$ for every $M \in \mathcal{M}_E^\epsilon$, property (3) holds. Combining Lemma B.1 and Lemma B.2 give us Lemma 4.7.

**Lemma B.1.** *Suppose that Algorithm 3 is called with the following inputs: a nonempty set of existing meta-actions $\mathcal{M}_E^\epsilon \subseteq \mathcal{M}^\epsilon$; $p_{M_I} \in \text{int}(P(M_I))$ for some $M_I \in \mathcal{M}_E^\epsilon$; $\epsilon \in (0,1)$ and $\rho \in (0,1)$. With probability at least $1 - (K + K^2/2)\rho$, Algorithm 3 returns either $\perp$, or a an action $a_{\widehat{M}} \in \widehat{M} \cap A^\star$ and a contract $p_{\widehat{M}} \in \text{int}(P(a_{\widehat{M}}))$, with $\widehat{M} \in \mathcal{M}^\epsilon \setminus \mathcal{M}_E^\epsilon$. Conditioned on the same event, the number of rounds required by the algorithm is at most $\frac{K}{\epsilon^2} \log(2d/\rho)$ in the first case, and at most $\frac{K^2}{2\epsilon^2} \log(2d/\rho)$ in the latter.*

*Proof.* Let us observe that the loop at Line 1 iterates over at most $K - 1$ actions. By Lemma 4.2 and an union bound, all queries at Line 5 are successful with probability at least $1 - (K - 1)\rho$, and account for less than:

$$N_1 := \frac{K}{\epsilon^2} \log\left(\frac{2d}{\rho}\right)$$

rounds. We now distinguish two cases, depending on wether Line 7 is executed or not.

If it is not executed, then it returns $\perp$ and terminates in at most $N_1$ rounds with probability at least $1 - (K - 1)\rho \geq 1 - (K + K^2/2)\rho$.

If Line 7 is executed, then let $\widehat{M}, p, M_1, p_{M_1}, \epsilon, \rho$ be the parameters given to Algorithm 4 at Line 7 Algorithm 3. We show that these parameters satisfy the assumptions of Lemma 4.8, conditioned on the event that all previous queries were successful. By construction, we have $\widehat{M} \notin \mathcal{M}_{\mathrm{E}}^\epsilon \ni M_1$. Furthermore, $p_{M_1} \in \mathrm{int}(P(M_1))$ by assumption. We also have that $p \in P(\widehat{M})$ as $\widehat{M}$ is computed by Algorithm 1 called over $p$ (Line 5 Algorithm 3). We can further observe that with probability one $p \in \mathrm{int}(P(\widehat{M}))$ and that segment between $p$ and $p_{M_1}$ never intersects multiple separating hyperplanes in a single point. Indeed, $p$ is sampled uniformly at random from a polytope $\mathcal{C}(a')$ (Line 4), and such a polytope has non-zero volume (Line 3). As a result, we can employ Lemma 4.8 and an union bound to show that, if Line 7 is executed, with probability at least $1 - (K + K^2/2)\rho$ the algorithm terminates in:

$$N_1 + \frac{K^2}{2\epsilon^2} \log\left(\frac{2d}{\rho}\right) \leq \frac{K^2}{2\epsilon^2} \log\left(\frac{2d}{\rho}\right)$$

rounds, and the algorithm returns a pair $(a_{\widehat{M}}, p_{\widehat{M}})$ with the correct properties. $\qquad\square$

**Lemma B.2.** *Suppose that Algorithm 3 is called with the following inputs: a nonempty set of existing meta-actions $\mathcal{M}_{\mathrm{E}}^\epsilon \subseteq \mathcal{M}^\epsilon$; a set $S := \{a_M\}_{M \in \mathcal{M}_{\mathrm{E}}^\epsilon}$ with $a_M$ being a representative action for $M \in \mathcal{M}_{\mathrm{E}}^\epsilon$; $p_{M_\mathrm{I}} \in \mathrm{int}(P(M_\mathrm{I}))$ for some $M_\mathrm{I} \in \mathcal{M}_{\mathrm{E}}^\epsilon$; $\epsilon \in (0, 1)$ and $\rho \in (0, 1)$. With probability at least $1 - K\rho$, if Algorithm 3 returns $\perp$, then the partition $\{\widetilde{P}(M)\}_{M \in \mathcal{M}^\epsilon}$ in Equation (1) is an $\epsilon$-partition.*

*Proof.* The loop at Line 1 iterates over at most $K - 1$ actions, executing a query at Line 5 at each iteration. By Lemma 4.2 and an union bound, all these queries are successful with probability at least $1 - (K - 1)\rho > 1 - K\rho$. We will continue the analysis under such an event and the assumption that Algorithm 3 returns $\perp$, and show that Equation 1 defines an $\epsilon$-partition. Notice that, for Algorithm 3 to return $\perp$, the if-statement at Line 6 is always evaluated to false, and Line 7 is never executed.

Take any $M \in \mathcal{M}_{\mathrm{E}}^\epsilon$ and $\bar{p} \in \widetilde{P}(M)$, where $\widetilde{P}(M)$ is the polytope defined by Equation 1. We show that, for every $a' \in A^\star$, it holds:

$$u^{\mathrm{A}}(\bar{p}, a_M) \geq u^{\mathrm{A}}(\bar{p}, a') - 2Kd\epsilon. \tag{5}$$

As $a(p) \in A^\star$, this proves that partition defined by Equation 1 is an $\epsilon$-partition.

Let us partition $A^\star$ into two sets, $A_{\mathrm{E}} := \bigcup_{M \in \mathcal{M}_{\mathrm{E}}} M \cap A^\star$ and $A_{\mathrm{C}} := A^\star \setminus A_{\mathrm{E}}$. According to Equation 1 and the definition of best-response region, we have that $u^{\mathrm{A}}(\bar{p}, a_M) \geq u^{\mathrm{A}}(\bar{p}, a_{M'})$. Applying Lemma 4.4 we get $u^{\mathrm{A}}(\bar{p}, a_{M'}) \geq u^{\mathrm{A}}(\bar{p}, a') - 2Kd\epsilon$. Hence, Equation 5 for the actions $a' \in A_{\mathrm{E}}$.

We thus have to prove Equation 5 for the actions $a' \in A_{\mathrm{C}}$. By construction, Algorithm 3 iterates over these actions (and some others that do not belong to $A^\star$). For every $a' \in A_{\mathrm{C}}$, we define the set of candidates

$$C(a') = \{\widetilde{p} \in \mathcal{P} \mid \forall M \in \mathcal{M}_{\mathrm{E}} \ u^{\mathrm{A}}(\widetilde{p}, a') \geq u^{\mathrm{A}}(\widetilde{p}, a_M) + 2Kd\epsilon\}, \tag{6}$$

as in Line 2 Algorithm 3. Fix an action $a' \in A_{\mathrm{C}}$. For Algorithm 3 to return $\perp$, there are two possibilities. The first is that $C(a')$ has null volume. The second is that there exists some $p \in \mathcal{C}(a')$ (sampled by Algorithm 3 at Line 4) such that $a(p) \in M$, with $M \in \mathcal{M}_{\mathrm{E}}^\epsilon$ under the event $\mathcal{E}$. Recall that we are analyzing the behavior of the algorithm when it correctly determines the meta-action $\widehat{M}$ of $a(p)$ (Line 5), and for it to return $\perp$, this meta-action must belong to $\mathcal{M}_{\mathrm{E}}^\epsilon$. We will analyze these two cases separately.

Suppose that $\mathrm{vol}(\mathcal{C}(a')) = 0$. The set $\mathcal{C}(a')$ is a polytope defined as the intersection of $\mathcal{P}$ with half-spaces $u^{\mathrm{A}}(\widetilde{p}, a') \geq u^{\mathrm{A}}(\widetilde{p}, a_M) + 2Kd\epsilon$, one for each $M \in \mathcal{M}_{\mathrm{E}}^\epsilon$. If it is empty or not fully dimensional, then for every $\widetilde{p} \in \mathcal{P}$ we have

$u^{\text{A}}(\widetilde{p}, a') \leq u^{\text{A}}(\widetilde{p}, a_M) + 2Kd\epsilon$ for at least one $M \in \mathcal{M}_{\text{E}}^{\epsilon}$. In particular, this holds for $\widetilde{p} = \bar{p} \in \mathcal{P}$, hence Equation 5 holds for $a'$.

The second case is that there exists some $p \in \mathcal{C}(a')$ (sampled by Algorithm 3 at Line 4) such that $a(p) \in M$, with $M \in \mathcal{M}_{\text{E}}^{\epsilon}$. We will show that this case cannot happen, as it leads to a contradiction, namely that $a' \notin A^{\star}$. First, we observe that $p \in \text{int}(\mathcal{C}(a'))$ with probability one, as it is sampled uniformly at random from a fully-dimensional polytope. Consequently, we have:

$$u^{\text{A}}(p, a') > u^{\text{A}}(p, a_M) + 2Kd\epsilon$$
$$\geq u^{\text{A}}(p, a(p)) + 2Kd\epsilon - 2Kd\epsilon = u^{\text{A}}(p, a(p)).$$

where the strict inequality holds according to Equation 6 and since $p$ is an interior point, while the last inequality holds thanks to Lemma 4.4, considering that $a(p), a_M \in M \cap A^{\star}$. This leads to a contradiction, $u^{\text{A}}(p, a') > u^{\text{A}}(p, a(p))$, where $a' \in A^{\star}$.

We have thus shown that Equation 5 holds for every $\widetilde{p} \in \mathcal{P}$ and every action $a' \in A^{\star}$, both those in $A_{\text{E}}$ and in $A_{\text{C}}$, with probability at least $1 - K\rho$. It follows that the cover defined by Equation 1 is an $\epsilon$-cover with such a probability. □

**Lemma 4.7.** *Suppose that Algorithm 3 is called with the following inputs: a nonempty set of existing meta-actions* $\mathcal{M}_{\text{E}}^{\epsilon} \subseteq \mathcal{M}^{\epsilon}$*; a set* $S \coloneqq \{a_M\}_{M \in \mathcal{M}_{\text{E}}^{\epsilon}}$ *with* $a_M \in M \cap A^{\star}$ *for every* $M \in \mathcal{M}_{\text{E}}^{\epsilon}$*; a contract* $p_{M_{\text{I}}} \in \text{int}(P(M_{\text{I}}))$ *for some* $M_{\text{I}} \in \mathcal{M}_{\text{E}}^{\epsilon}$*;* $\epsilon, \rho \in (0, 1)$*. Then, Algorithm 3 correctly terminates (according to Definition 4.6) with probability at least* $1 - (2K + K^2/2)\rho$*, by using* $\frac{K^2}{\epsilon^2} \log \frac{2d}{\rho}$ *rounds.*

*Proof.* The parameters provided in input to Algorithm 3 satisfy the assumptions of both Lemma B.1 and Lemma B.2. Applying these two lemmas and an union bound proves the statement. □

### B.3. Find Representative Action

Algorithm 4 is responsible for finding an existing action $a_{M_1} \in A^{\star} \cap M_1$ and a contract $p_{M_1} \in \text{int}(P(M_1))$, given in input two contracts $p_1 \in \text{int}(P(M_1))$ and $p_2 \in \text{int}(P(M_2))$, with $M_1 \neq M_2$. In order to do so, it considers the segments $L$ between $p_1$ and $p_2$ and iterates over the contracts $p^{\circ}$ at the intersections of $L$ with some hyperplane $H_{a,a'}$, with $a \in M_1$ and $a' \notin M_1$. Given a $p^{\circ}$, it computes two contracts $p, p' \in L$ using the procedure `ComputeClosePoints`, which we describe in the following. The pseudocode is provided in Algorithm 7.

---

**Algorithm 7** `ComputeClosePoints`

---

**Require:** contracts $p_1 \neq p_2 \in \mathcal{P}$, actions $a \neq a' \in A$
1: Let $L$ be the segments between $p_1$ and $p_2$
2: $p^{\circ} \leftarrow H_{a,a'} \cap L$
3: $w_1 \leftarrow p_1 - p^{\circ}$
4: $\epsilon_1 \leftarrow \min\{\epsilon > 0 \mid \exists\{i, j\} \neq \{a, a'\} : p^{\circ} + \epsilon w_1 \in H_{i,j} \vee \epsilon = 1\}$
5: $\epsilon \leftarrow \epsilon_1/2$
6: $p \leftarrow p^{\circ} + \epsilon w_1$
7: $w_2 \leftarrow p_2 - p^{\circ}$
8: $\epsilon_2 \leftarrow \min\{\epsilon > 0 \mid \exists\{i, j\} \neq \{a, a'\} : p^{\circ} + \epsilon w_2 \in H_{i,j} \vee \epsilon = 1\}$
9: $\epsilon' \leftarrow \epsilon_2/2$
10: $p' \leftarrow p^{\circ} + \epsilon' w_2$
11: **Return** $p, p'$

---

The goal of Algorithm 7 is to find $p, p' \in L$ such that the segment $L' \subset L$ between them intersects only the separating hyperplane $H_{a,a'}$ and no other. To do so, the two points must be close to $p^{\circ}$, that is the point at the intersection of $L$ with $H_{a,a'}$. Notice that $p$ and $p'$ lie on opposite sides of $H_{a,a'}$. Since we know all the possible separating hyperplanes, we can simply enumerate them and look at where they intersect $L$. Let $p^{\circ} + \epsilon_1 w_1$ be the closest intersection on one side of $H_{a,a'}$, with $w_1 = p_1 - p^{\circ}$ and $\epsilon_1$ an appropriate number strictly larger than zero. Similarly, let $p^{\circ} + \epsilon_2 w_2$ be the closest intersection of these separating hyperplanes on the other side of $H_{a,a'}$, with $w_2 = p_2 - p^{\circ}$. Algorithm 7 has simply to compute any $p = p^{\circ} + \epsilon w_1$ and $p' = p^{\circ} + \epsilon' w_2$, with $\epsilon' < \epsilon_1$ and $\epsilon' < \epsilon_2$. We complete the code by taking into account the case where no separating hyperplane intersects $L$ in one or both sides of $H_{a,a'}$.

The result is formalized as follows.

**Lemma B.3.** *Let $p_1, p_2 \in \mathcal{P}$ be two different contracts. Suppose that the segment $L$ between them interests $H_{a,a'}$ in a point $p^\circ \notin \{p_1, p_2\}$, and that it does not lie on any separating hyperplane. Then Algorithm 7 computes two contracts, $p, p' \in L \setminus \{p^\circ\}$, such that the segment between them intersects separating hyperplanes only in $p^\circ$. Furthermore, $p$ and $p'$ belong to the interior of some (possibly the same) best-response regions.*

*Proof.* Let $L_1$ be the segment between $p^\circ$ and $p_1$, and $L_2$ the segment between $p^\circ$ and $p_2$. Let also $p, p'$ be the two contracts computed by Algorithm 7, and let $S_1$ be the segment between $p$ and $p^\circ$, and $S_2$ the segment between $p'$ and $p^\circ$. We argue that Algorithm 7 compute a point $p \in L_1 \setminus p^\circ$ such that $S_1 \setminus \{p^\circ\}$ does not intersect any separating hyperplanes.

Let $w_1 = p_1 - p^\circ$, so that the segment $L_1$ consists of the contracts $x = p^\circ + \epsilon w_1$ for $\epsilon \in [0, 1]$. Algorithm 7 computes $\epsilon_1 \in (0, 1]$ defined as:

$$\epsilon_1 = \min\{\epsilon > 0 | \exists\{i, j\} p^\circ + \epsilon w_1 \in H_{i,j} \vee \epsilon = 1\}.$$

Subsequently, it computes $p = p^\circ + w_1 \epsilon_1 / 2$. Suppose that $L_1 \setminus \{p^\circ\}$ does not intersect any separating hyperplane. The same must hold for $S_1 \setminus \{p^\circ\} \subseteq L_1 \setminus \{p^\circ\}$.

Suppose instead that $L_1 \setminus \{p^\circ\}$ intersects at least one separating hyperplane. We observe that $L_1 \subseteq L$ does not lie on any separating hyperplane. Therefore, by construction Algorithm 7 computes $\epsilon_1 \in (0, 1]$ such that $S_1 \setminus \{p^\circ\} \subseteq L_1 \setminus \{p^\circ\}$ does not intersects any separating hyperplane.

With a similar argument, we can show that $S_2 \setminus \{p^\circ\}$ does not intersects any separating hyperplane. As a result, the segment $S_1 \cup S_2$ between $p$ and $p'$ intersects hyperplanes only in $p^\circ$. Finally, since $\epsilon_1 \in (0, 1]$, we have that $p \in S_1 \setminus \{p^\circ\}$. Therefore, $p$ does not belong to any separating hyperplane, and is thus in the interior of some best-response region. The same must hold for $p' \in S_2 \setminus \{p^\circ\}$, concluding the proof. $\qquad\square$

Finally, we can provide the proof of Lemma 4.8. Intuitively, Algorithm 4 queries the contracts $p, p'$, discovering the meta-actions $M$ and $M'$, respectively. If they are different, $M \neq M'$, then it must be the case that $a(p) = a$ and $a(p') = a'$. Indeed, $H_{a,a'}$ is the single separating hyperplane between $p$ and $p'$. Since $a \in M_1$, the algorithm has found the required pair $a_{M_1} = a$ and $p_{M_1} = p$. Furthermore, at least one iteration of Algorithm 4 finds $M \neq M'$, because $L$ traverses both $P(M_1)$ and $P(M_2)$. Therefore, one iteration of Algorithm 4 considers $p^\circ$ at the boundary of $P(M_1)$ and finds $M \neq M'$.

**Lemma 4.8.** *Suppose that Algorithm 4 is called with the following inputs: two different meta-actions $M_1 \neq M_2 \in \mathcal{M}^\epsilon$; two contracts $p_1 \in \text{int}(P(M_1))$ and $p_2 \in \text{int}(P(M_2))$; $\epsilon, \rho \in (0, 1)$. Moreover, assume that the line segment connecting $p_1$ and $p_2$ never intersects multiple separating hyperplanes in a single point. Then, Algorithm 4 computes $a_{M_1} \in A^\star \cap M_1$ and $p_{M_1} \in \text{int}(P(a_{M_1}))$ by using at most $\frac{K^2}{2\epsilon^2} \log \frac{2d}{\rho}$ rounds, with probability at least $1 - \rho K^2 / 2$.*

*Proof.* We let $L$ be the line segment between $p_1$ and $p_2$. Such a segment starts in the interior of the best-response region $P(M_1)$ and ends in the interior of $P(M_2)$. Furthermore, it does not intersects multiple hyperplanes in any single point.

Let $\mathcal{I}$ be the set of contracts defined as:

$$\mathcal{I} := \{p^\circ \in L \mid p^\circ \in L \cap H_{a,a'}, a \in M_1, a' \notin M_1\}. \tag{7}$$

This set contains a finite number of contracts. In particular, it contains at most one contract for each pair of actions $a \in M_1, a' \notin M_1$. Therefore:

$$|\mathcal{I}| \leq |M_1||A \setminus M_1| = |M_1|(K - |M_1|) \leq \frac{K^2}{4}$$

As Algorithm 4 iterates over the set $\mathcal{I}$ and calls Algorithm 1 two times for iteration, it follows that it executes at most $K^2/2$ queries. Employing Lemma 4.2 and an union bound, with probability at least $1 - \rho K^2 / 2$ all these queries are successful and the algorithm requires:

$$\frac{K^2}{2\epsilon^2} \log \left( \frac{2d}{\rho} \right)$$

rounds. We will continue the analysis conditioned on the event that all the queries executed by the algorithm returned the correct meta-actions.

Consider a generic iteration of Algorithm 4. Let $p^\circ \in \mathcal{I}$ be the contract of such an iteration, and let $a \in M_1, a' \notin M_1$ be the unique actions such that $p^\circ \in H_{a,a'}$. We observe that $p^\circ \in \text{int}(\mathcal{P})$, as both $p_1$ and $p_2$ do not lie on the boundary of the search space. Thus, according to Lemma B.3, at Line 7 Algorithm 4 computes two contracts, $p, p' \in L$, such that the line segment between them intersects only the separating hyperplane $H_{a,a'}$. Furthermore, both $p$ and $p'$ belong do the interior of some (possibly the same) best-response regions, *i.e.* $p \in \text{int}(P(a(p)))$ and $p' \in \text{int}(P(a(p')))$. Under the event $\mathcal{E}$, Algorithm 4 finds the two meta actions $M, M' \in \mathcal{M}^\epsilon$ such that $a(p) \in M$ and $a(p') \in M'$.

Algorithm 4 computes the pair $(a_{M_1}, p_{M_1})$ (Line 10) when the If-statement at Line 9 is evaluated as true, *i.e.*, when $M \neq M'$. We argue that when this happens, it holds that $a_{M_1} \in A^\star \cap M_1$ and $p_{M_1} \in \text{int}(P(a_{M_2}))$. The algorithm computes $a_{M_1} = a$ and $p_{M_1} = p$. Since $a \in M_1$ and $p \in \text{int}(P(a(p)))$, we can prove that the pair is consistent by showing that $a(p) = a$.

Suppose, by contradiction, that $a(p) \neq a$. It follows that $a(p) = a(p')$, as there is no separating hyperplane in the form $H_{a(p),\cdot}$ between the two contracts. Indeed, the single separating hyperplane between the two contracts is $H_{a,a'}$. However, we also have that $a(p) \in M$ and $a(p') \in M'$, with $M = M'$. Since different meta-action are disjoint sets of actions, this implies that $a(p) \neq a(p)$, reaching a contradiction.

In order to conclude the proof, we need to prove that there exist an iteration where the if-statement al Line 9 is evaluated as true. Let $\widetilde{\mathcal{I}} := L \cap \partial P(M_1)$, where $\partial P(M_1)$ is the boundary of $P(M_1)$. We observe that $\widetilde{\mathcal{I}} \subseteq \mathcal{I}$. To see this, take a contract $p \in L \cap \partial P(M_1)$. Since $p \in \partial P(M_1)$, it either belongs to $\partial \mathcal{P}$, or it lies on a hyperplane in the form $H_{a,a'}$, with $a \in M_1, a' \notin M_1$. The first case is not possible, as $L$ is the segment between two contracts in the interior of $\mathcal{P}$, and $p \in L$. Therefore, $p \in L \cap \partial P(M_1)$ belongs to $L \cap H_{a,a'}$. By Equation 7, this a contract in $\mathcal{I}$. As a result, Algorithm 4 iterates over the contracts in $\widetilde{\mathcal{I}} \subseteq \mathcal{I}$, as it iterates over $\mathcal{I}$. We will now show that for any $p^\circ \in \widetilde{\mathcal{I}}$, the if-statement at Line 9 is evaluated true, and that $|\widetilde{\mathcal{I}}| \geq 1$.

As the the segment $L$ must traverse the boundary of $P(M_1)$ to go from $p_1 \in P(M_1)$ to $p_2 \notin P(M_1)$, the cardinality of $\widetilde{\mathcal{I}}$ cannot be zero. Let $p^\circ \in \widetilde{\mathcal{I}}$ and and observe that $p^\circ \in H_{a,a'}$, with $a \in M_1, a' \notin M_1$, while it does not belong to any other separating hyperplane. Thus, it must be the case that the points $p, p'$ computed by Algorithm 4 are such that $p \in \text{int}(P(a))$ and $p' \in \text{int}(P(a'))$. It follows that the if-statement al Line 9, as $a$ and $a'$ belong to different $\epsilon$-meta-actions. Thus, there exist an iteration where Algorithm 4 correctly computes the pair $(a_{M_1}, p_{M_1})$. □

## B.4. Initial point

---
**Algorithm 8** FindInitialPoint
---
**Require:** $\epsilon, \rho \in (0,1)$
 1: $p_1 \leftarrow$ sample a point uniformly at random from $\mathcal{P}$.
 2: $M_1 \leftarrow \texttt{Query}(p_1, \epsilon, \rho)$
 3: **for** $a_1 \in M_1$ **do**
 4:     **if** $\texttt{TestCover}(\{M_1\}, \{a_1\}, p_1, \epsilon, \rho)$ returns a pair $(a_{M_I}, p_{M_I})$ **then**
 5:         **Return** $(a_{M_I}, p_{M_I})$
 6:     **end if**
 7: **end for**
 8: **Return** the partition defined by Equation (8)
---

Algorithm 8 is the first subprocedure employed by Algorithm 6. It's goal is to find, if possible, an initial existing action $a_{M_I}$ for some $M_I \in \mathcal{M}^\epsilon$, together with some $p_{M_I} \in \text{int}(P(M_I))$. These can then be used as starting points by Algorithm 2.

First, we need to acknowledge that there are instances where finding any existing action employing Algorithm 1 to interact with the agent is not possible. For instance, if $A^\star \subseteq M$, for some meta-action $M \in \mathcal{M}_E^\epsilon$, then any possible query would return $M$, and therefore it would not be possible to determine an actual existing action by means of any number of queries. However, in such a case we can simply provides an $\epsilon$-partition consisting of the single region $\widetilde{P}(M) = \mathcal{P}$, and skip Algorithm 2 completely. As a result, we will devise an algorithm that either finds an initial pair $(a_{M_I}, p_{M_I})$, or returns a cover defined as:

$$\widetilde{P}(M) = \begin{cases} \mathcal{P} & M = M_1, \\ \emptyset & \text{otherwise,} \end{cases} \tag{8}$$

for some $M_1 \in \mathcal{M}^\epsilon$.

To attain this goal, Algorithm 8 queries a contract $p_1$ sampled uniformly at random from $\mathcal{P}$, discovering some meta action $M_1 \in \mathcal{M}^\epsilon$. Notice that this contract belongs to the interior of its meta-best-response region almost surely.

We observe that Equation 8 is equivalent to Equation 1 when $\mathcal{M}_\mathrm{E}^\epsilon = \{M_1\}$ and $a_{M_1} = a(p_1)$. In other words, if Algorithm 8 were to know $a(p_1)$, it could employ `TestPartition` to verify wether $M_1$ alone provides the desired partition. Thus, Algorithm 8 tries to "guess" the action $a(p_1) \in M_1$. For every possible candidate $a_1 \in M_1$, it employs `TestPartition` to verify wether $\widetilde{P}(M_1) = \mathcal{P}$ correctly approximate the entire contract space. Notably, if every test is successful (including the one with the correct guess $a_1 = a(p_1)$), then Algorithm 8 can conclude that Equation 8 is an $\epsilon$-partition

Instead, if at least one test fails, then `TestPartition` has found a consistent pair $(a_{M_\mathrm{I}}, p_{M_\mathrm{I}})$ for some $M_\mathrm{I} \in \mathcal{M}^\epsilon$. This consistent pair can thus be employed as a starting point for Algorithm 2.

This analysis leads to the following lemma.

**Lemma B.4.** *Given in input $\epsilon, \rho \in (0,1)$ and with probability at least $1 - (3K + 2K^2 + K^3/2)\rho$, Algorithm 8 satisfies the following properties:*

1. *it terminates in $\frac{2K^2}{\epsilon^2} \log\left(\frac{2d}{\rho}\right)$ rounds.*

2. *it returns either an $\epsilon$-cover, or an action $a_{M_\mathrm{I}} \in M \cap A^\star$ and a contract $p_{M_\mathrm{I}} \in \mathrm{int}(P(M_\mathrm{I}))$ for some $M_\mathrm{I} \in \mathcal{M}^\epsilon$.*

*Proof.* Let $p_1$ be the contract sampled at random at Line 1 Algorithm 8, and $M_1$ the action returned by the query at Line 2. For the sake of the analysis, we define the following events. First, we let $\mathcal{E}_1$ be the event under which $p_1 \in \mathrm{int}(P(M_1))$ and the first query takes less than $\frac{1}{\epsilon^2} \log(2d/\rho)$ rounds. Second, we let $\mathcal{E}_2$ be the event under which the loop of Algorithm 8 terminates in $\frac{3K^2}{2\epsilon^2} \log(2d/\rho)$ rounds at most. Third, we let $\mathcal{E}_3$ be the event under which the algorithm executes a correct test according to Definition 4.6 using the guess $a_1 = a(p)$, or it never reaches the iteration with the correct guess $a_1 = a(p)$. Fourth, we let $\mathcal{E}_4$ be the event under which either the algorithm returns a pair $(a_{M_\mathrm{I}}, p_{M_\mathrm{I}})$ satisfying the statement, or the loop ends without executing Line 5.

The event $\mathcal{E}_1$ has probability at least $1 - \rho$, as $p_1$ belongs to the interior of a best-response region with probability one, and the query is successful with probability at least $1 - \rho$ by Lemma 4.2.

We also argue that, conditioned on the event $\mathcal{E}_1$, the event $\mathcal{E}_2$ has probability at least $1 - (K^2 + K^3/2)\rho$. Algorithm 8 iterates over every action in $M_1$, for a total of at most $K$ iterations. Conditioned on the event $\mathcal{E}_1$, whenever Algorithm 8 employs Algorithm 3, the assumptions of Lemma B.1 are satisfied. By applying Lemma B.1 and an union bound, we have that, with probability at least $1 - (K^2 + K^3/2)\rho$, the loop terminates in:

$$(K-1)\frac{K}{\epsilon^2} \log\left(\frac{2d}{\rho}\right) + \frac{K^2}{2\epsilon^2} \log\left(\frac{2d}{\rho}\right) \le \frac{3K^2}{2\epsilon^2} \log\left(\frac{2d}{\rho}\right)$$

rounds, accounting for at most a single execution that returns a pair $(a_{M_\mathrm{I}}, P_{M_\mathrm{I}})$ and terminates the loop early. As a result, we have $\mathbb{P}(\mathcal{E}_1 | \mathcal{E}_2) \ge 1 - (K^2 + K^3/2)\rho$.

Next, we argue that, conditioned on $\mathcal{E}_1$ and $\mathcal{E}_2$, event $\mathcal{E}_3$ has probability at least $1 - (2K + K^2/2)\rho$. Indeed, if the iteration with the correct guess is reached, then the test of such iteration is correctly executed with probability at least $1 - (2K + K^2/2)\rho$ (Lemma 4.7). Otherwise, if such an iteration is not reached, then by definition the event happens.

Finally, we argue that, conditioned on $\mathcal{E}_1$ and $\mathcal{E}_2$, event $\mathcal{E}_4$ has probability at least $1 - (K + K^2/2)\rho$. If the loop ends, then the event happens by definition. Otherwise, we employ Lemma B.1 at the last iteration of the loop to show that the event has probability at least $1 - (K + K^2/2)\rho$.

Finally, by an union we notice that the intersection of the four events has probability at least:

$$1 - \left(1 + K^2 + \frac{K^3}{2} + 2K + \frac{K^2}{2} + K + \frac{K^2}{2}\right)\rho \ge 1 - \left(3K + 2K^2 + \frac{K^3}{2}\right)\rho,$$

and, conditioned on these events, the algorithm number of rounds required by the algorithm is at most:

$$\frac{1}{\epsilon^2} \log\left(\frac{2d}{\rho}\right) + \frac{3K^2}{2\epsilon^2} \log\left(\frac{2d}{\rho}\right) = \frac{3K^2 + 2}{2\epsilon^2} \log\left(\frac{2d}{\rho}\right) \le \frac{2K^2}{\epsilon^2} \log\left(\frac{2d}{\rho}\right),$$

after which it returns either a cover or an action $a_{M_{\mathrm{I}}} \in M \cap A^\star$ and a contract $p_{M_{\mathrm{I}}} \in \mathrm{int}(P(M_{\mathrm{I}}))$ for some $M_{\mathrm{I}} \in \mathcal{M}^\epsilon$. $\square$

# C. Proofs omitted from Section 5

**Lemma C.1.** *Let* $a, a' \in M$, *where* $M$ *is an* $\epsilon$-*meta-action. For every contract* $p \in \mathcal{P}$ *it holds that:*

$$u^P(p, a') - Kd\epsilon \leq u^P(p, a) \leq u^P(p, a') + Kd\epsilon$$

*Proof.* Since $a, a'$ belong to the same meta-action, we have that $\|F_a - F_{a'}\|_\infty \leq K\epsilon$. Consequently we can upper bound $|u^P(p, a) - u^P(p, a')|$ as:

$$
\begin{aligned}
|u^P(p, a) - u^P(p, a')| &= |\langle F_a - F_{a'}, r - p \rangle| \\
&\leq \sum_{\omega \in \Omega} |(F_{a,\omega} - F_{a',\omega})(r_\omega - p_\omega)| \\
&\leq \sum_{\omega \in \Omega} K\epsilon 1 \\
&= Kd\epsilon.
\end{aligned}
$$

This proves the statement. $\qquad\square$

**Lemma 5.1.** *Suppose that Algorithm 5 is called with inputs an* $\epsilon$-*partition* $\{\widetilde{P}(M)\}_{M \in \mathcal{M}^\epsilon}$ *and a set* $\{a_M\}_{M \in \mathcal{M}^\epsilon}$ *with* $a_M \in M$ *for every* $M \in \mathcal{M}^\epsilon$. *Then:*

$$u^P(\widetilde{p}, a_{\widetilde{M}}) \geq \mathrm{OPT} - Kd\epsilon - 2\sqrt{2Kd\epsilon}.$$

*Proof.* Let $p^\star$ be an optimal contract and $p^\diamond := (1 - \sqrt{CKd\epsilon})p^\star + r\sqrt{2Kd\epsilon}$. We argue that the following holds:

$$u^P(p^\diamond, a_{M^\diamond}) \geq \mathrm{OPT} - Kd\epsilon - 2\sqrt{2Kd\epsilon}, \tag{9}$$

where $M^\diamond \in \mathcal{M}^\epsilon$ is a meta-action such that $p^\diamond \in \widetilde{P}(M^\diamond)$.

Since $\{\widetilde{P}(M)\}_{M \in \mathcal{M}^\epsilon}$ is an $\epsilon$-cover, there exists an action $a^\diamond \in M^\diamond \cap A^\star$ such that:

$$u^A(p^\diamond, a^\diamond) \geq u^A(p^\diamond, a(p)) - 2Kd\epsilon.$$

We recall that in general $a_{M^\diamond} \neq a^\diamond$. The first is used by Algorithm 5 as a representative of the distribution of $M$, while the latter is an approximate best response in $p^\diamond$. It may also be the case that $a_{M^\diamond} \notin A^\star$, while it always hold that $a^\diamond \in A^\star$. By the definition of agent's utility function, we observe that:

$$
\begin{aligned}
\langle F_{a^\diamond}, p^\diamond \rangle - c_{a^\diamond} &\geq \langle F_{a(p^\diamond)}, p^\diamond \rangle - c_{a(p^\diamond)} - 2Kd\epsilon \\
&\geq \langle F_{a(p^\star)}, p^\diamond \rangle - c_{a(p^\star)} - 2Kd\epsilon,
\end{aligned}
$$

where the last inequality holds because $a(p^\diamond)$ is the best response in $p^\diamond$. Rearranging and plugging in the definition of $p^\diamond$:

$$
\begin{aligned}
2Kd\epsilon &\geq \langle F_{a(p^\star)} - F_{a^\diamond}, p^\diamond \rangle - c_{a(p^\star)} + c_{a^\diamond} \\
&= \langle F_{a(p^\star)} - F_{a^\diamond}, (1 - \sqrt{2Kd\epsilon})p^\star + r\sqrt{2Kd\epsilon} \rangle - c_{a(p^\star)} + c_{a^\diamond} \\
&= \sqrt{2Kd\epsilon}\langle F_{a(p^\star)} - F_{a^\diamond}, r - p^\star \rangle + \langle F_{a(p^\star)} - F_{a^\diamond}, p^\star \rangle - c_{a(p^\star)} + c_{a^\diamond} \\
&= \sqrt{2Kd\epsilon}\langle F_{a(p^\star)} - F_{a^\diamond}, r - p^\star \rangle + u^A(p^\star, a(p^\star)) - u^A(p^\star, a^\diamond) \\
&\geq \sqrt{2Kd\epsilon}\langle F_{a(p^\star)} - F_{a^\diamond}, r - p^\star \rangle = \sqrt{2Kd\epsilon}(\mathrm{OPT} - u^P(p^\star, a^\diamond)).
\end{aligned}
$$

where the last inequality holds due to the optimality of $a(p^\star)$, formally that $u^A(p^\star, a(p^\star)) \geq u^A(p^\star, a^\diamond)$. Therefore:

$$u^P(p^\star, a^\diamond) \geq \mathrm{OPT} - \sqrt{2Kd\epsilon}. \tag{10}$$

We can now lower bound $u^{\mathrm{P}}(p^\diamond, a_{M^\diamond})$ as follows:

$$u^{\mathrm{P}}(p^\diamond, a_{M^\diamond}) \geq u^{\mathrm{P}}(p^\diamond, a^\diamond) - Kd\epsilon \tag{11a}$$

$$= \langle F_{a^\diamond}, r - (1 - \sqrt{2Kd\epsilon})p^\star - r\sqrt{2Kd\epsilon}\rangle - Kd\epsilon \tag{11b}$$

$$= \langle F_{a^\diamond}, r - p^\star\rangle + \sqrt{2Kd\epsilon}\langle F_{a^\diamond}, p^\star - r\rangle - Kd\epsilon \tag{11c}$$

$$\geq \langle F_{a^\diamond}, r - p^\star\rangle - \sqrt{2Kd\epsilon} - Kd\epsilon \tag{11d}$$

$$= u^{\mathrm{P}}(p^\star, a^\diamond) - \sqrt{2Kd\epsilon} - Kd\epsilon \tag{11e}$$

$$\geq \mathrm{OPT} - 2\sqrt{2Kd\epsilon} - Kd\epsilon. \tag{11f}$$

where Equation 11a holds thanks to Lemma C.1, Equation 11b by the definition of $p^\diamond$, Equation 11d as $\|r\|_\infty \leq 1$ and $\sum_{\omega \in \Omega} F_{a^\diamond, \omega} = 1$, and Equation 11f by Equation 10. As a result, Equation (9) holds.

Finally, we conclude the proof by observing that $u^{\mathrm{P}}(\widetilde{p}, a_{\widetilde{M}}) \geq u^{\mathrm{P}}(p^\diamond, a_{M^\diamond})$ by construction (Line 1 Algorithm 5). $\qquad\square$

**Lemma 5.2.** *Suppose that Algorithm 5 is called with inputs an $\epsilon$-partition $\{\widetilde{P}(M)\}_{M \in \mathcal{M}^\epsilon}$ and a set $\{a_M\}_{M \in \mathcal{M}^\epsilon}$ with $a_M \in M$ for every $M \in \mathcal{M}^\epsilon$. Then:*

$$u^P(\widetilde{p}^\diamond) \geq \mathrm{OPT} - (4\sqrt{2} + 2)Kd\sqrt{\epsilon}.$$

*Proof.* Let $\widetilde{p}$ and $\widetilde{M}$ the contract and meta-action computed by Algorithm 5 at Line 1, formally defined by:

$$\widetilde{p}, \widetilde{M} := \arg\max_{\substack{M \in \mathcal{M}^\epsilon \\ p \in \widetilde{P}(M)}} \langle r - p, F_{a_M}\rangle.$$

The algorithm returns the contract $\widetilde{p}^\diamond := \left(1 - \sqrt{2Kd\epsilon}\right)\widetilde{p} + r\sqrt{2Kd\epsilon}$. By Definition 4.1, there exists some action $\widetilde{a} \in \widetilde{M} \cap A^\star$ that is an approximate best-response in $\widetilde{p}$, formally:

$$u^{\mathrm{A}}(\widetilde{p}, \widetilde{a}) \geq u^{\mathrm{A}}(\widetilde{p}, a(p)) - 2Kd\epsilon.$$

Therefore, by applying Proposition 2.4 by (Dutting et al., 2021) we have:

$$u^{\mathrm{P}}(\widetilde{p}^\diamond) \geq u^{\mathrm{P}}(\widetilde{p}, \widetilde{a}) - 2\sqrt{2Kd\epsilon}.$$

A a result:

$$u^{\mathrm{P}}(\widetilde{p}^\diamond) \geq u^{\mathrm{P}}(\widetilde{p}, \widetilde{a}) - 2\sqrt{2Kd\epsilon}$$
$$\geq u^{\mathrm{P}}(\widetilde{p}, a_{\widetilde{M}}) - 2\sqrt{2Kd\epsilon} - Kd\epsilon$$
$$\geq \mathrm{OPT} - 2Kd\epsilon - 4\sqrt{2Kd\epsilon}$$
$$\geq \mathrm{OPT} - (4\sqrt{2} + 2)Kd\sqrt{\epsilon},$$

which holds by Lemma C.1, Lemma 5.1, and the fact that $\sqrt{\epsilon} > \epsilon$ and $Kd > \sqrt{Kd}$. $\qquad\square$

