# OpenReview forum: "Online Contract Design With Unknown Technology"
_ICML.cc/2026/Conference — ICML 2026 regular_

### Official Review · Reviewer_dE7z · 2026-03-04

**Soundness:** 2
**Presentation:** 3
**Significance:** 1
**Originality:** 2
**Overall Recommendation:** 4
**Confidence:** 3

**Summary:**

The paper studies the setting of online Principal-Agent problems and tackles the problem of handling the exponential dependence on the size of the problem instance. The assumption on the agent's behavior is to be incentive compatible and individually rational. The goal of the Principal is that the agent select favorable IC-IR actions (among a finite set). To model the incentive compatible aspect, a best-response region is defined: for any agent's action $a$, it is the set of contracts such that  the action is incentive compatible for the contract (it is a polytope). The formulation as a bandit arises naturally with the arms seen as a subset of possible actions. To improve over this naive approach, the method first learns a partition such that in the partition, the agent's IC-IR action is close from it. It leads to improved regret bounds.

**Compliance With Llm Reviewing Policy:**

Affirmed.

**Final Justification:**

The rebuttal answered my concerns.

**Key Questions For Authors:**

Questions:
- Would such a method be robust to an agent that would be approximately best-responding (for instance picking a random action with a small probability)?
- Do you have an intuition of how far the regret bound is from a lower bound?
- Would the setup be interesting to translate in the case of one principal and multiple agents (e.g. [5], which by the way could also be a relevant reference)?

[5] Dütting, P., Ezra, T., Feldman, M., & Kesselheim, T. (2023, June). Multi-agent contracts. In Proceedings of the 55th Annual ACM Symposium on Theory of Computing (pp. 1311-1324).

**Limitations:**

Yes

**Strengths And Weaknesses:**

Strengths:
- The paper nicely introduces the principal-agent problem in a formal way. It is overall rigorous.
- It contributes to research in that field, regarding the dependence of the regret on the number of arms.

Weaknesses:
- I think that the major weakness of the paper is its incrementality, I do not believe that the studied problem extremely interesting nor motivating.
- The presentation of the proofs could be improved, typically, "B. Proofs and Algorithms omitted from Section 4" is very verbose and requires improvement.
- Adding a conclusion may be nice.

Suggestions:
- I understand that "technology" in the title ("the set of actions truly available to the agent") has been used by [1] but I find it a bit misleading in this context, coming from a more ML background.
- For the setup considered, I find the following references quite relevant: [2] for the aspect of a principal interacting with a learning agent, [3] for the bandit/repeated aspect of the principal-agent problem and [4] which is a major reference regarding principal-agent problems.
- Graphs and drawings would help the visualization of the geometric aspects.

[1] Carroll, G. (2015). Robustness and linear contracts. American Economic Review, 105(2), 536-563.

[2] Guruganesh, G., Kolumbus, Y., Schneider, J., Talgam-Cohen, I., Vlatakis-Gkaragkounis, E. V., Wang, J. R., & Weinberg, S. M. (2024). Contracting with a learning agent. arXiv preprint arXiv:2401.16198.

[3] Scheid, A., Tiapkin, D., Boursier, E., Capitaine, A., Mhamdi, E. M. E., Moulines, É., ... & Durmus, A. (2024). Incentivized learning in principal-agent bandit games. arXiv preprint arXiv:2403.03811.

[4] Laffont, J. J., & Martimort, D. (2002). The theory of incentives: the principal-agent model. Princeton university press.

---

> ### Author Rebuttal · Authors · 2026-03-31
>
> We thank the Reviewer for the effort put in reviewing our paper. We provide detailed responses to the Reviewer's concerns and questions below.
>
> > I think that the major weakness of the paper is its incrementality, I do not believe that the studied problem extremely interesting nor motivating.
>
> We kindly disagree with the Reviewer on this point. Our paper studies a principal-agent model in which a set of candidate agent actions is known to the principal beforehand. This model was neglected by previous works, even though it captures several relevant emerging application scenarios (e.g., machine learning task delegation) and, as we show in our work, makes it possible to attain better regret guarantees. The algorithm we developed is based on iteratively searching for truly existing actions in this candidate set. It compares the known actions with the candidate ones to determine whether it should look for new existing actions, and if that is the case, it again analyzes the candidates to determine where and how to find them. This leads to an algorithm that is fundamentally different from existing ones, as they did not have the concept of candidate actions to begin with.
>
> > The presentation of the proofs could be improved, typically, "B. Proofs and Algorithms omitted from Section 4" is very verbose and requires improvement.
>
> We will surely extend the proof exposition in the final version of the paper.
>
> > Adding a conclusion may be nice.
>
> We agree with the Reviewer, we will add a proper "Conclusions" section in the final version of the paper.
>
> > I understand that "technology" in the title ("the set of actions truly available to the agent") has been used by [1] but I find it a bit misleading in this context, coming from a more ML background.
>
> We agree with the Reviewer that the term "technology" could be misleading for readers with an ML background, even though it is technically correct. We are open to change the title and replace the term "technology" with, for example, "set of candidate actions" throughout the paper.
>
> > For the setup considered, I find the following references quite relevant: [2] for the aspect of a principal interacting with a learning agent, [3] for the bandit/repeated aspect of the principal-agent problem and [4] which is a major reference regarding principal-agent problems.
>
> We thank the Reviewer for suggesting these references, we will discuss them in the final version of the paper.
>
> > Would such a method be robust to an agent that would be approximately best-responding (for instance picking a random action with a small probability)?
>
> The method can be slightly modified to work in the case mentioned by the Reviewer. It is sufficient to increase the value of $T_{\epsilon,\rho}$ in Algorithm 1 to make the queries robust to approximate best-responses. Specifically, we need $T_{\epsilon,\rho} = \Theta(\frac{1}{(1-\delta)^2\epsilon^2}\log(\frac{d}{\rho}))$, where $\delta$ is the probability that the agent picks an action at random.
>
> > Do you have an intuition of how far the regret bound is from a lower bound?
>
> Deriving a tight lower bound in this setting remains an interesting open problem. We observe that any lower bound in our setting would also hold for the scenario considered by Zhu et al. (2023) and Bacchiocchi et al. (2024), in which deriving a tight lower bound is still an open problem. Our result matches the dependency on $T$ by Bacchiocchi et al. (2024), while it has polynomial dependency on the other input parameters.
>
> > Would the setup be interesting to translate in the case of one principal and multiple agents (e.g. [5], which by the way could also be a relevant reference)?
>
> It would be interesting to study online contract design with multiple agents, but such a step is out of the scope of our paper. The structure of these problems is fundamentally different from the single-agent setting, as they involve complex agent interactions and combinatorial structures.
>
> For instance, Dutting et al. (2023) consider a scenario with linear contracts and binary actions, and even in this simple setting an optimal contract can only be approximated up to a constant factor in polynomial time. Instead, in our single-agent problem the agent can pick one of multiple actions, and the optimal non-linear contract can be computed in polynomial time given full knowledge of the game.
>
> [5] Dütting, P., Ezra, T., Feldman, M., & Kesselheim, T. (2023, June). Multi-agent contracts. In Proceedings of the 55th Annual ACM Symposium on Theory of Computing (pp. 1311-1324).

---

> > ### Author Rebuttal · Reviewer_dE7z · 2026-04-02
> >
> > I thank the authors for their rebuttal, which answers my questions and concerns and I increased my score accordingly. I still think that "technology" is not extremely appropriate in the title but it's a minor issue.

---

### Official Review · Reviewer_YT28 · 2026-03-12

**Soundness:** 3
**Presentation:** 4
**Significance:** 3
**Originality:** 3
**Overall Recommendation:** 5
**Confidence:** 3

**Summary:**

This paper considers the online learning framework for hidden-action principal-agent problems. While prior work has established sublinear regret bounds for this setting, existing algorithms typically suffer from an exponential dependence on the size of the problem instance, such as the number of outcomes or the agent's action space.

Concretely, in the authors' model, the principal does not know the agent's actual technology (the set of truly available actions $A^*$) but knows a superset $A$ of candidate actions. The paper establishes a regret upper bound of $\mathcal{O}(K^{7/5} d^{4/5} T^{4/5} \log(d K^3 / \delta))$, where $K$ is the size of the superset $A$ and $d$ is the number of outcomes. This achieves a regret that scales polynomially in the instance size parameters while remaining sublinear in the number of interaction rounds $T$.

From a technical perspective, the main idea is to cluster candidate actions into "meta-actions" to group actions with similar outcome distributions. The authors iteratively learn an $\epsilon$-partition of the contract space. They do this by identifying existing meta-actions and finding representative "proxy" actions. The algorithm then leverages this approximate partition to compute and commit to an approximately optimal contract.

**Compliance With Llm Reviewing Policy:**

Affirmed.

**Final Justification:**

I appreciate the authors' rebuttal. My questions have been sufficiently answered, and I remain supportive of the paper’s acceptance.

**Key Questions For Authors:**

1. The time horizon dependence $T^{4/5}$ is worse than optimal bounds seen in simpler linear settings.
2. Is there any negative lower bound result?

**Limitations:**

yes

**Strengths And Weaknesses:**

Strengths

I find the problem setting reasonable and well-motivated. Online contract design is an active research area. The assumption that the principal knows a superset of possible actions—rather than the exact feasible set—is highly practical for real-world delegation tasks, such as in machine learning. I would also like to highlight the quality of the algorithmic design: circumventing the exponential dimensionality barrier by clustering indistinguishable actions is an elegant solution.

Weaknesses

See the questions.

---

> ### Author Rebuttal · Authors · 2026-03-31
>
> We thank the Reviewer for the insightful comments and for positively evaluating our work.
>
> > The time horizon dependence $T^{4/5}$ is worse than optimal bounds seen in simpler linear settings.
>
> Indeed, when we restrict to only linear contracts it is possible to achieve $\widetilde{\mathcal{O}}(\sqrt{nT})$ regret without even knowing a superset of possible actions (Bacchiocchi et al., 2025a). This is because the setting becomes one-dimensional and the leader's utility function is piecewise linear and monotonically decreasing in each piece, and this allows to employ different techniques that achieve better regret rates. Such techniques cannot be employed in the general setting, where best-response regions are high-dimensional polytopes with possibly exponentially many vertices and we are interested in the actual distributions induced by the actions, rather than simply their expected reward.
>
> > Is there any negative lower bound result?
>
> Deriving a tight lower bound in this setting remains an interesting open problem. We observe that any lower bound in our setting would also hold for the scenario considered by Zhu et al. (2023) and Bacchiocchi et al. (2024), in which deriving a tight lower bound is still an open problem. Our result matches the dependency on $T$ by Bacchiocchi et al. (2024), while it has polynomial dependency on the other input parameters.

---

> > ### Author Rebuttal · Reviewer_YT28 · 2026-04-06
> >
> > I appreciate the authors' rebuttal. My questions have been sufficiently answered, and I remain supportive of the paper’s acceptance.

---

### Official Review · Reviewer_GGdX · 2026-03-13

**Soundness:** 3
**Presentation:** 4
**Significance:** 3
**Originality:** 3
**Overall Recommendation:** 4
**Confidence:** 3

**Summary:**

The paper studies a contracting problem between a principal and agent where the principal does not know the exact set of actions, but rather knows a superset of the agent's action space. For every action in the superset, the principal knows the distribution it induces over outcomes and costs. The only thing the principal does not know if the action is accessible to the agent. So they propose an algorithm where the principal learns which actions are accessible by matching empirical distribution of deploying a contract with actions whose true distribution is similar. This is possible since the principal knows the distribution over outcomes induced by various actions. Their algorithm achieves sublinear regret (T^{4/5}) relative to optimal contract and has polynomial dependence on all problem parameters.

**Compliance With Llm Reviewing Policy:**

Affirmed.

**Key Questions For Authors:**

-

**Limitations:**

yes

**Strengths And Weaknesses:**

Strengths:
1. The problem is well-motivated
2. They produce a learning algorithm with sublinear regret and polynomial dependence on problem parameters.

Weaknesses:
The framing of the paper is slightly misleading. The paper seems to imply that it is getting over some exponential dependence from previous work and is doing so by introducing a model where principal has uncertainty about exact subset of actions available to the agent. But this is confusing because it seems like adding an additional form of uncertainty which should make the principal's problem harder.

What is enabling the better rates is that they move to an easier problem setting where the principal has full information about costs and outcome distributions. In previous work, the principal has to learn these distributions and costs. If we remove this uncertainty, there is nothing left to learn. And so this paper introduces a different, potentially easier source of uncertainty. So it is not fair to compare the rates of this work with rates of past work.

---

> ### Author Rebuttal · Authors · 2026-03-31
>
> We thank the Reviewer for the insightful comments.
>
> > The framing of the paper is slightly misleading. The paper seems to imply that it is getting over some exponential dependence from previous work and is doing so by introducing a model where principal has uncertainty about exact subset of actions available to the agent. But this is confusing because it seems like adding an additional form of uncertainty which should make the principal's problem harder. What is enabling the better rates is that they move to an easier problem setting where the principal has full information about costs and outcome distributions. In previous work, the principal has to learn these distributions and costs. If we remove this uncertainty, there is nothing left to learn. And so this paper introduces a different, potentially easier source of uncertainty. So it is not fair to compare the rates of this work with rates of past work.
>
> In both our setting and those of previous works, there is uncertainty over the costs and distributions associated with the agent actions. However, while in previous works the principal does not have any idea about the possible costs and distributions, in our model the principal knows which are the possible candidates. In this sense, we are not introducing additional uncertainty, but rather reducing it, which is what enables better regret guarantees. We apologize for the misunderstanding; we will improve the presentation in the final version of the paper.
>
>
> There are no questions, has the Reviewer forgotten to add them?

---

> > ### Author Rebuttal · Reviewer_GGdX · 2026-04-01
> >
> > I think it will be beneficial to clarify more the difference in uncertainty models between this work and previous work and also clarify what this means for comparing upper and lower bounds in both settings

---

### Official Review · Reviewer_p7fd · 2026-03-13

**Soundness:** 3
**Presentation:** 4
**Significance:** 3
**Originality:** 3
**Overall Recommendation:** 4
**Confidence:** 3

**Summary:**

This paper studies the contract design problem through the lens of online learning, in which a principal repeatedly interacts with an agent to learn the optimal contract. Existing online learning algorithms for this problem achieve regret bounds that grow sublinearly with the number of interaction rounds. However, they suffer from an exponential dependence on the input size, particularly on the number of outcomes. To address it, the paper makes the assumption that the principal has partial prior information about the agent’s possible actions, costs, and outcome distributions and derives the regret bound that is polynomial in the number of candidate actions and outcomes.

**Compliance With Llm Reviewing Policy:**

Affirmed.

**Final Justification:**

Thank the authors for their effort. I will maintain my positve score and look forward to seeing the revised version with discussion regarding the lower bounds.

**Key Questions For Authors:**

1. Regarding the key assumptions in the paper, the assumption that the principal has access to a prior superset of the agent’s possible actions seems relatively mild. However, the assumption that the principal knows all candidate outcome distributions and costs exactly appears considerably stronger. What happens if this information is only approximate rather than exact? Would the proposed approach still work in that setting, and if so, what changes would be required in the algorithm or the analysis?

2. Are there any lower bounds for the setting with unknown technology (under the assumptions considered in the paper)? (It might be challenging, can the authors give some intuition regarding the barrier?)

**Limitations:**

No. It would be more helpful if the authors discussed the limitations of the assumptions underlying both the model and the algorithms.

**Strengths And Weaknesses:**

**Strengths**
1. The model studied in this paper is intriguing, and the technical contributions appear strong. In particular, the introduction of partial-information assumptions into contract design, the use of clustering ideas for meta-actions, and the construction of the partition are all quite novel.
2. The results looks significant. By incorporating additional prior information about the agent’s actions, the paper avoids the exponential dependence on the input size and achieves sublinear regret with only polynomial dependence.
3. The paper is well-presented, especially the high-level ideas for design algorithms and explanation for each sub-routine.

**Weaknesses**
1. Some of the assumptions look a little bid demanding.
2. There is no empricial result in the paper. For example, missing experimental results to validate that the learned $\epsilon$-partitions approximate true meta-best-response regions in practice.
3. There is no lower bound provided in the paper.

---

> ### Author Rebuttal · Authors · 2026-03-31
>
> We thank the Reviewer for the insightful comments.
>
> > 1. Regarding the key assumptions in the paper, the assumption that the principal has access to a prior superset of the agent’s possible actions seems relatively mild. However, the assumption that the principal knows all candidate outcome distributions and costs exactly appears considerably stronger. What happens if this information is only approximate rather than exact? Would the proposed approach still work in that setting, and if so, what changes would be required in the algorithm or the analysis?
>
> The problem of achieving no-regret when only approximate information on candidate actions is available is an interesting problem that may be addressed in a future work. We conjecture that our method may be adapted to such a setting if the information is sufficiently precise. The main challenge involves Algorithm 4, which needs to query additional contracts in the neighborhood of $p^\circ$. The set $C(a)$ in Algorithm 3 and the number of rounds of Algorithm 1 should also change to account for the uncertainty.
>
> However, as the precision of the information reduces, our approach becomes useless, as the candidate set of actions is no longer useful. Finding the minimum precision (in terms of $T$) required to achieve no-regret remains an open problem.
>
>
> > 2. Are there any lower bounds for the setting with unknown technology (under the assumptions considered in the paper)? (It might be challenging, can the authors give some intuition regarding the barrier?)
>
> There is no known lower bound for this setting. We observe that any lower bound in our setting would also hold for the scenario considered by Zhu et al. (2023) and Bacchiocchi et al. (2024), in which deriving a tight lower bound is still an open problem. As in Bacchiocchi et al. (2024), one of the main challenges is to understand which sort of approximation of the best-response regions can be learned, and what contract can be computed given such an approximation. In our setting with a known set of candidate actions, our algorithm improves the dependency on $K$, while still matching the dependency of Bacchiocchi et al. (2024) in $T$. Whether this can be improved or not remains an open problem.

---

> > ### Author Rebuttal · Reviewer_p7fd · 2026-04-01
> >
> > Thank the authors for their responses. I appreciate the clarifications. For the second question, I have some follow-up questions.
> >
> > 1. In the rebuttal, you write: “We observe that any lower bound in our setting would also hold for the scenario considered by Zhu et al. (2023) and Bacchiocchi et al. (2024), in which deriving a tight lower bound is still an open problem.”. I don't think this statements logically supports that having such a lower bound for your model is also hard, since it looks like your model is a special case of the setting in Zhu et al. (2023). Also, I checked Zhu et al. (2023), where they do provide a lower-bound result. Have you tried to adapt or extend that lower-bound construction to your setting? If so, what is the main technical barrier?
> >
> > 2. I agree deriving a tight lower bound would be very hard, however, i presume that there should be at least some discussion about the lower bound or impossibility result for the model to support your algorithm, or has a discussion paragraph of why deriving one is hard, and what kinds of barriers arise in this setting.
> >
> > Please correct me if I I have misunderstood anything from your rebuttal. Thank you.

---

> > > ### Author Response · Authors · 2026-04-02
> > >
> > > We thank the Reviewer for the response and the insightful comment. We will add a paragraph discussing the problem of deriving a tight lower and the relationship with the lower bound by Zhu et al. (2023) in the final version of the paper.
> > >
> > > > Have you tried to adapt or extend that lower-bound construction to your setting?
> > >
> > > Zhu et al. (2023) devise an algorithm that achieve $\mathcal{O}(\sqrt{d}T^{\frac{2d}{2d+1}})$ regret, independently of the number of agent's actions. They then provide a lower bound of $\Omega(T^{\frac{d+1}{d+2}})$ by using instances with $\Theta(T^{\frac{d}{d+2}})>T^{\frac{1}{2}}$ actions. Our goal is instead to exploit the information on the candidate actions to attain better guarantees, hence we expect something dependent on $K$ in the regret bound---specifically, we would like to attain $\mathcal{O}{(\text{poly}(K,d)T^C)}$ for some constant $C<1$. Considering a regime where $K \ge T^{\frac{1}{2}}$ is not really interesting, as even an algorithm that attains $\mathcal{O}(K\sqrt{T})$ would incur in linear regret in this case. Instead, our algorithm aims at achieving small regret when $K$ is small with respect to $T$.
> > >
> > > > Why deriving a tight lower bound would be very hard?
> > >
> > > Given that we want an algorithm that attains $\mathcal{O}{(\text{poly}(K,d)T^C)}$ regret, we believe that a lower bound on the achievable constant $C$ would be the most significative option. To this end, we observe that a naive approach (Section 3.1) attains $\mathcal{O}(2^K\sqrt{T})$ regret. We conjecture that removing the exponential dependency on $K$ necessarily increases the exponent of $T$. Indeed, our algorithm attains $\mathcal{O}{(\text{poly}(K,d)T^{\frac{4}{5}})}$. Determining whether such a dependency on $T$ is tight or not is a challenging open problem. It involves determining what sort of approximation of the best-response regions can be learnt while keeping the regret polynomial in $K$. We can only observe that the lower bound would employ instances with $K$ of the order of $\log(T)$ candidate actions, so that any algorithm whose regret is exponential in $K$ would incur in linear regret. The structure that such instances should have is not clear at all.
> > >
> > > Finally, let us also observe that Bacchiocchi et al. (2024) incur in $\mathcal{O}(T^{4/5})$ regret when all other parameters are constant, and they do not have a lower bound as well. We agree with the Reviewer that this does not directly imply that a lower bound in our setting is hard. However, since our problem and that of Bacchiocchi et al. (2024) are intimately connected, we believe that their lower bounds should be based on very similar instances. Specifically, we expect that our additional assumption allows us to achieve polynomial regret in the instance size, but the same dependency on $T$ attained by Bacchiocchi et al. (2024).

---

### Decision · Program_Chairs · 2026-04-30

**Decision:**

Accept (regular)

**Comment:**

The paper studies online contract design where the principal knows a superset of the agent's possible actions but not which are truly available, and proposes an algorithm achieving T^{4/5} regret with polynomial dependence on all problem parameters. All four reviewers are positive, finding the problem well-motivated and the algorithmic ideas intuitive. The main open question is the lack of matching lower bounds, which the authors discussed thoughtfully. I recommend acceptance.